# Outlier Suppression+: Accurate quantization of large language models by equivalent and effective shifting and scaling

**Xiuying Wei[1,2,3] , Yunchen Zhang[2,5] , Yuhang Li[4] , Xiangguo Zhang[2],**
**Ruihao Gong[1,2]\*, Jinyang Guo[1] , Xianglong Liu[1]**

[1]State Key Lab of Software Development Environment, Beihang University
[2]SenseTime Research [3]School of Computer and Communication Sciences, EPFL
[4]Yale University, [5]UESTC

xiuying.wei@epfl.ch, yuhang.li@yale.edu, {jinyangguo, xlliu}@buaa.edu.cn

{zhangyunchen, zhangxiangguo, gongruihao}@sensetime.com

## Abstract

Post-training quantization (PTQ) of transformer language models faces significant challenges due to the existence of detrimental outliers in activations. We observe that these outliers are concentrated in specific channels and are asymmetric across channels. To address this issue, we propose the Outlier Suppression+ (OS+) framework, which contains the channel-wise shifting for asymmetry and channel-wise scaling for concentration. We show that these operations can be seamlessly migrated into subsequent modules while maintaining equivalence. Second, we propose a fast and stable scheme to calculate effective shifting and scaling values. The channel-wise shifting aligns the center of each channel for removal of outlier asymmetry. The channel-wise scaling quantitatively evaluates changes brought by migration and quantization for better quantization burden balance. We validate our OS+ under both standard and fine-grained quantization settings with models including BERT, OPT, BLOOM, BLOOMZ, and LLaMA. Comprehensive results across various tasks demonstrate the superiority of our approach. Especially, with standard quantization, OS+ can achieve near-floating-point performance on both small models and large language models on 8-bit and 6-bit. Besides, we establish a new state-of-the-art for 4-bit BERT with 15.5% improvement. Our code is available at https://github.com/ModelTC/Outlier_Suppression_Plus.

## 1 Introduction

Transformer language models (e.g., BERT, LLMs) have garnered significant attention due to their remarkable performance and scalable model size. These models have evolved from hundreds of millions of parameters (Devlin et al., 2018; Liu et al., 2019; Radford et al., 2018) to hundreds of billions of parameters (Brown et al., 2020; Zhang et al., 2022; Smith et al., 2022). This necessitates the employment of compression techniques (Han et al., 2015; Hinton et al., 2015; Zoph and Le, 2016; Le-Cun et al., 1989) for practical deployment. Among these techniques, quantization (Jacob et al., 2018) has emerged as a general and primary paradigm for reducing both memory footprint and computation overhead.

However, quantization, particularly post-training quantization (Choukroun et al., 2019; Banner et al., 2018; Wu et al., 2020) under the setting of limited data and GPU resources, has become increasingly challenging on these models (e.g., a 12% accuracy drop in BERT (Bondarenko et al., 2021) and catastrophic degradation in OPT-175B (Dettmers et al., 2022)). This is caused by the presence of detrimental outliers in activation (e.g., the range of distribution can be 80 in BERT and even 140 in OPTs), which prevents discrete numbers from accurately representing continuous ones.

To combat the bottleneck, researchers make in-depth investigations and find that outliers mainly concentrate on certain channels. Some works (Bondarenko et al., 2021; Dettmers et al., 2022) suggest fine-grained quantization schemes and offer extra bit levels for outlier channels. Others (Wei et al., 2022b; Xiao et al., 2022) take the activation scaling to scale outliers and migrate scaling values to subsequent weights for FP equivalence. However, the former might hurt the quantization acceleration effect while the latter determines scaling values without the consideration of minimizing the change introduced by migration and quantization, which we find is sub-optimal. Meanwhile, we also identify a new outlier characteristic that previous works overlooked but is also responsible for the large tensor range.

In this paper, we propose the Outlier Suppression+ framework composed of channel-wise shift-

---

*Corresponding author.

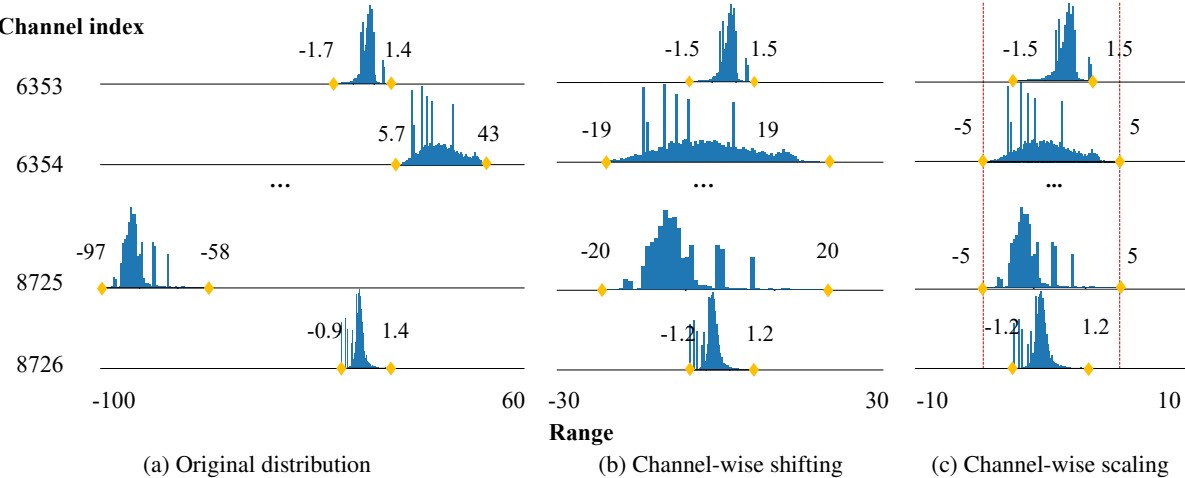

Figure 1: Distribution of OPT-66B. Fig. 1a shows the original distribution with asymmetric outliers consistently occurs at certain channels, owning considerable range (-97, 43). Fig. 1b depicts the channel-wise shifting operation to decrease the tensor range by eliminating the asymmetry. Fig. 1c further scales down the outliers to threshold 5 and finally results in a distribution ranging from -5 to 5.

ing and scaling to effectively pursue better quantization performance while equivalently keeping the FP output. First, we find a new feature of outliers that they stay in asymmetric shape across channels (e.g., in Fig. 1a, one problematic channel on OPT-66B occupies the negative axis from -97 to -58 while another one has positive values ranging from 5.7 to 43). This outlier asymmetric presentation could cause a significantly wide distribution of tensor like 140 even composed of channels with relatively small ranges like 39. Thus, we propose the channel-wise shifting operation, which shifts the activation across channels to eliminate the impact of asymmetry. Together with channel-wise scaling for concentrated outliers, a unified migration pattern is introduced to seamlessly transfer the reversed effects of these operations to later modules to maintain equivalent FP models. Second, we devise deliberate schemes to determine effective shifting and scaling values. The shifting vector aligns the center of each channel, reducing the whole tensor range to its maximum channel range. The scaling values quantitatively minimize the interactive output change of the activation and weights induced by migration and quantization, achieving a balanced quantization burden with a fast and stable search procedure.

Our algorithm can be carried out efficiently and enjoy affordability on real hardware, producing more quantization-friendly models in minutes and requiring no extra inference burden on LLMs. To this end, our main contributions can be summarized into three aspects:

1. We find a new feature of outliers that show asymmetric shapes across channels and then propose the channel-wise shifting operation, along with taking channel-wise scaling for the outlier concentration attribute. A unified migration pattern that migrates their reversed effects to later modules is designed to guarantee an equivalent FP network.

2. We propose fast and stable ways to determine effective shifting and scaling values. Shifting values eliminate the asymmetry feature across channels while scaling values scale down outlier channels towards a quantitative optimization objective.

3. We assess the efficacy of our approach under both standard and fine-grained quantization settings. On standard one, OS+ achieves near-floating-point performance on 8-bit and 6-bit BERT, OPTs, BLOOM, and BLOOMZ. On fine-grained one, OS+ can surpass others by 9.41% on 4-bit LLaMA with per-token quantization and obtain lossless results on 4-bit OPT with per-group quantization.

## 2 Related work

Due to the space limit, we give the most relevant papers here and put a complete related work in the Appendix A. In the realm of PTQ, researchers have discovered that the poor performance of transformer language models should be attributed to extreme outliers in activations, which exhibit special characteristics from both channel and token aspects. Thus, we will introduce related works

from the two aspects.

**Channel aspect.** Outliers consistently emerge in certain channels over different inputs. Bondarenko et al. (2021) employs a per-embedding-group quantization scheme that uses different quantization parameters for distinct channel groups, while Dettmers et al. (2022) suggests utilizing FP16 representations for problematic channels holding signals over 6. Wei et al. (2022b) introduces an outlier suppression (OS) framework with one of components called Gamma Migration. Observing that outliers accumulate in certain channels, it adopts a scaling vector to scale outliers and migrates it to subsequent modules. Xiao et al. (2022) further proposes calculating scaling values by equalizing ranges between activations and weights and evaluates on large language models. Guo et al. (2023) discards normal values adjacent to outliers, making room for outliers with customized GPU support. To consider the standard quantization, we find that Wei et al. (2022b) and Xiao et al. (2022) still waste a large portion of quantization levels on the extreme outlier asymmetry across channels. Meanwhile, Wei et al. (2022b) simply views the scaling parameter in LayerNorm (LN) as the scaling vector for outliers, which might not always be consistent with the outlier distribution. Xiao et al. (2022) that adopts the heuristic way and obtains equalized ranges between activation and weights lacks quantitative evaluation of their output change induced by migration and quantization.

**Token aspect.** Different tokens exhibit varying degrees of outliers. Dettmers et al. (2022); Yao et al. (2022) introduce a novel scheme called per-token quantization that dynamically computes quantization parameters for each token. Wei et al. (2022b) investigates the clipping impact of outliers and recommends finding an appropriate clipping range in a token-wise manner. In this paper, we focus on the channel aspect and might combine these techniques when necessary.

## 3 Preliminary

**Basic Notations.** We denote matrices as upper case letters (e.g., $X$) and vectors as lower case letters (e.g., $x$). Operator $\odot$ and $\oslash$ represent element-wise multiplication and division for matrices or vectors. We use $WX$ as matrix-matrix multiplication. Furthermore, $X_{t,j}$ refers to the element of the $t$-th token and the $j$-th channel in transformer models. $Q(\cdot)$ denotes the quantization function.

**Quantization.** We indicate standard quantization as per-tensor activation quantization, per-channel, or per-tensor weight quantization here because such schemes will not separate the integer matrix multiplication. Per-tensor means assigns quantization parameters for each tensor and per-channel for each output channel. Also, for some fine-grained ways, we mainly consider per-token (Yao et al., 2022) and per-group (Yao et al., 2023) here, which calculates quantization parameters in each token or group.

## 4 Method

We first present our equivalent shifting and scaling operations, then introduce ways to determine effective values for them.

### 4.1 Equivalent shifting and scaling

In this section, we comprehensively investigate outlier features, naturally introducing the design of shifting and scaling operations, followed by a unified migration pattern.

#### 4.1.1 Outlier shifting and scaling

**Channel-wise shifting.** For transformers, especially LLMs, we find that outliers show asymmetric behavior among channels. Recall that in Fig. 1a, the 8725-th channel displays a hard negative interval (-97, -58), while another channel dominates a positive one (5.7, 43). Due to this asymmetry, even if the range of each channel is relatively small, such as 40 and 39 for outlier channels and minuscule values for normal channels, the range of the entire tensor can swell to a considerably large value (e.g., 140, ranging from -97 to 43), which negatively affects quantization performance.

To handle this issue, we propose channel-wise shifting, which can eliminate the impact of asymmetry by taking the following operation:

$$\widetilde{X'} = X - z, \qquad (1)$$

where $z$ serves as a row vector ($z \in \mathbb{R}^n$) and shifts the activation for each channel. In this way, with a carefully designed $z$ which we will introduce in Sec. 4.2.1, the new tensor $\widetilde{X'}$ can get rid of the outlier asymmetry attribute. For example, by aligning the centers of each channel in Fig. 1b, the range can be reduced to 40 (the maximum channel range) from 140 (the large tensor range). Finally, note that this operation is not the conventional shifting operation for symmetric quantization, as it operates channel-wisely and provides better distribution for per-tensor quantization.

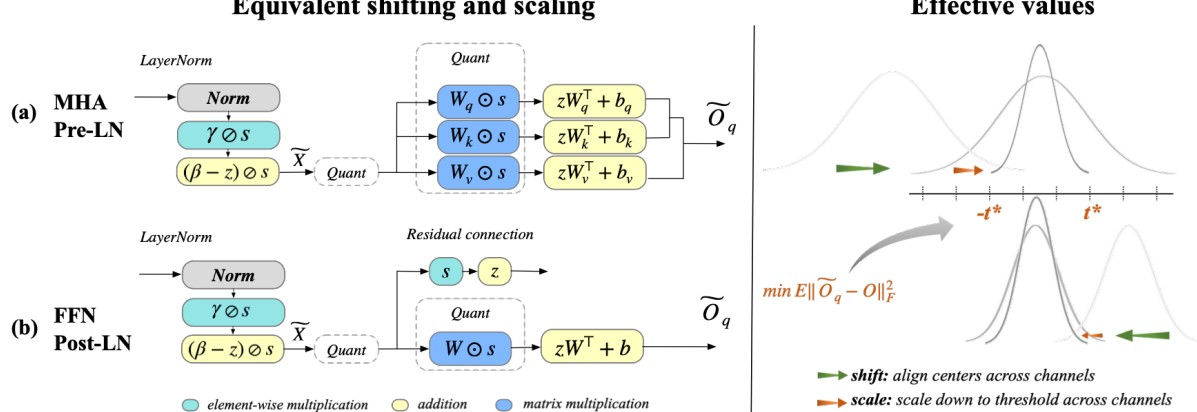

**Figure 2: Left**: We show the equivalent shifting and scaling operations by giving two representative examples: (a) for problematic output of Pre-LN (LayerNorm put inside residual connection) with Multi-Head Attention (MHA) structure; (b) for problematic output of Post-LN (LayerNorm put before residual connection) with Feed-Forward Network (FFN). **Right**: For effective shifting and scaling values, the shifting vector can align the center of each channel to 0 and the scaling vector would shrink outliers into the outlier threshold $t$ which is searched based on its left metric.

**Channel-wise scaling.** Apart from the asymmetry feature across channels, there also exists the outlier concentration phenomenon (Wei et al., 2022b) that outliers predominantly accumulate in specific channels over various inputs. For example, the 8725-th and the 6354-th channels in Fig. 1a hold more aggressive values than others. Therefore, after shifting, we equip with the channel-wise scaling to narrow them down to further alleviate the quantization difficulty.

$$\widetilde{X} = (X - z) \oslash s. \qquad (2)$$

In the above equation, the row vector $s \in \mathbb{R}^n$ scales the shifted tensor for each channel and brings final quantization-friendly activation $\widetilde{X}$. For example, in Fig. 1c, a tensor with a size of 10 can be obtained if we scale down channels with signals over 5. Detailed calculation of $s$ will be given in Sec. 4.2.2.

**Implementation.** It is easy to implement these operations. Take the output of LayerNorm Fig. 2 as an example, we only need to replace its linear transformation parameters $\beta$ and $\gamma$ with $(\beta - z) \oslash s$ and $\gamma \oslash s$ to achieve shifting and scaling effects. For others, we can update parameters in the former DeQuant function.

### 4.1.2 Unified migration pattern

As mentioned in Eq. (1) and Eq. (2), we subtract $z$ and divide $s$ to make the problematic activation resilient to quantization. To keep an equivalent FP model, a unified migration pattern is proposed that transfers both reversed shifting and scaling vectors to subsequent modules. We demonstrate

the feasibility of this algorithm on two common structures.

**Linear Layer.** First, we consider a prevalent scenario where a linear (convolutional) layer immediately follows. Reversing the above operations (i.e., $(\widetilde{X} \odot s + z)W^\top + b$) equals to updating the $W \in \mathbb{R}^{m,n}$ and $b \in \mathbb{R}^m$ in the next layer, given by

$$
\begin{aligned}
& (\widetilde{X} \odot s + z)W^\top + b \\
=\ & (\widetilde{X} \odot s)W^\top + zW^\top + b \qquad (3) \\
=\ & \widetilde{X}(W^\top \odot s^\top) + (zW^\top + b).
\end{aligned}
$$

According to Eq. (3), weight and bias can absorb $s$ and $z$, respectively, and thus becomes:

$$
\widetilde{W} = W \odot \begin{bmatrix} s_1 & s_2 & \dots & s_n \\ s_1 & s_2 & \dots & s_n \\ \dots \\ s_1 & s_2 & \dots & s_n \end{bmatrix},
$$
$$
\widetilde{b} = zW^\top + b. \qquad (4)
$$

For example, Fig. 2(a) depicts the typical challenging activation (output of LayerNorm) in the attention structure, all following weights and biases can absorb the shifting and scaling signals without any extra computation burden.

**Residual connection.** Second, we consider the case where a residual connection is applied after the LayerNorm structure (Post-LN) and fed into the quantized input. As shown in Fig. 2b, in addition to linear layer transformation, the identity function will be substituted with channel-wise multiplication and addition to maintain equivalence. We demonstrate that these increased calculations

will only incur a negligible inference burden in Sec. 5.5.

Finally, because $s$ and $z$ serve as shared parameters across tokens and batches of data, the unified migration pattern can be well-implemented and produce the same output without additional computation most of the time.

## 4.2 Effective shifting and scaling

Based on the equivalent shifting and scaling operations, in this section, we propose a fast and stable scheme to pursue effective values.

### 4.2.1 Shifting values

The design of the shifting vector should eliminate the impact of asymmetry across channels. Thus, we devise to align the center of each channel to 0 so that the outlier channel will not occupy only the positive or negative side. In detail, $z$ is defined as the average of the minimum and maximum signals in each channel, given by:

$$z_j = \frac{\max(\boldsymbol{X}_{:,j}) + \min(\boldsymbol{X}_{:,j})}{2}, \qquad (5)$$

With the channel-wise shifting now, the tensor range reduces to the largest channel range, getting rid of being defined by asymmetric outliers.

### 4.2.2 Scaling values

The design of the scaling vector should further scale down outliers while bringing marginal impact on following weight quantization. The following parts introduce how to obtain it with the proposed optimization objective and procedure.

**Challenges.** Recall that the equivalent transformation Eq. (4) also scales weights and potentially leads to inferior weight quantization, which requires us to calculate elaborate scaling values to reach a quantization balance between activation and weights. Nevertheless, we find previous works (Wei et al., 2022b; Xiao et al., 2022) either ignore the affected following weight or take a heuristic way that simply equalizes ranges of activation and weights. Unlike them, we think the key point is to minimize their interactive output change resulting from migration and quantization (a detailed analysis is available in Table 6). Hence, a new optimization objective is proposed.

**Optimization objective.** We first study the simple case that the problematic activation acts as the input of one linear layer (e.g., Fig. 2b). Instead of minimizing quantization errors

of activation and weight separately (i.e., $\min_{\boldsymbol{s}} \mathbb{E}\left[\|Q((\boldsymbol{X} - \boldsymbol{z}) \oslash \boldsymbol{s}) - (\boldsymbol{X} - \boldsymbol{z}) \oslash \boldsymbol{s}\|_F^2\right]$ and $\min_{\boldsymbol{s}} \mathbb{E}\left[\|Q(\boldsymbol{W} \odot \boldsymbol{s}) - \boldsymbol{W} \odot \boldsymbol{s}\|_F^2\right]$), a task loss perspective is adopted by concerning their matrix multiplication output. We measure the output change after scaling and quantizing weight and activation to pursue effective factors, given by:

$$\min_{\boldsymbol{s}} \mathbb{E}[\| \underbrace{Q((\boldsymbol{X} - \boldsymbol{z}) \oslash \boldsymbol{s}) Q(\boldsymbol{W} \odot \boldsymbol{s})^\top + \widetilde{\boldsymbol{b}}}_{\text{output after scaling and quantization}} - \underbrace{(\boldsymbol{X}\boldsymbol{W}^\top + \boldsymbol{b})}_{\text{original FP output}} \|_F^2], \qquad (6)$$

where the mean squared error (MSE) is used to quantify the difference.

*Multiple linear layers:* Furthermore, we study the case for multiple linear layers like the attention structure (Fig. 2a), where three weights will be multiplied by the same scaling vector and calculated with the same suppressed activation.

In this scenario, their matrix multiplication outputs produced by scaled and quantized matrices are marked as $\widetilde{\boldsymbol{Q}}_q, \widetilde{\boldsymbol{K}}_q, \widetilde{\boldsymbol{V}}_q$, (Original outputs are denoted as $\boldsymbol{Q}, \boldsymbol{K}, \boldsymbol{V}$). Applying Eq. (6) to three linear layers separately and simply summing the losses can make it difficult to illustrate their different importance and usages. Therefore, we employ the attention mechanism as a post-process function to reasonably organize their scaling and quantization information, given by:

$$\min_{\boldsymbol{s}} \mathbb{E}[\|\text{softmax}(\widetilde{\boldsymbol{Q}}_q \widetilde{\boldsymbol{K}}_q^\top) \widetilde{\boldsymbol{V}}_q - \text{softmax}(\boldsymbol{Q}\boldsymbol{K}^\top)\boldsymbol{V}\|_F^2]. \qquad (7)$$

Normalization and masking are omitted for notation simplicity, and it can be seen that information from the first two linear layers has been encapsulated within the attention map.

**Optimization procedure.** Toward the above objective, a fast and stable procedure is introduced to search the scaling vector. First, we find that scaling down only channels with outliers can bring better performance. Because channels with normal activations can exhibit more variation over different inputs, it can be difficult to find a decent scaling value for them. Also, considering that they are not responsible for low quantization performance, scaling them is not necessary. Second, we propose to optimize an alternate variable called outlier threshold $t$, which would squeeze only channels with an activation range over $t$ into $(-t, t)$ and keep others intact (Fig. 2). Essentially, $t$ here is used to specify

which channel to scale down, the final scaled activation range, as well as the scaling values in the following weights.

This technique simplifies the complex problem with numerous variables $s$ to a single variable $t$. Then we adopt the simple grid search for $t$ to minimize the objective Eq. (6), Eq. (7). After getting the effective $t$, the scaling vector is calculated as:

$$s_j = \max(1.0, \frac{\max(\boldsymbol{X}_{:,j} - \boldsymbol{z}_j)}{t}). \qquad (8)$$

## 5 Experiments

The evaluations are designed to show: **I.** satisfactory predictions of our OS+ for both small and large language models with standard quantization; **II.** consistent performance of OS+ on even lower-bit with fine-grained quantization; **III.** ablation study; **III.** analysis like computation complexity.

### 5.1 Set up

**Quantization setting.** Both the standard and fine-grained quantization are considered. For the standard one, we take quantization nodes the same as in Wei et al. (2022b); NVIDIA (2022), always adopt per-tensor activation quantization, consider per-tensor (fastest speed) and per-channel (high performance) weight quantization. For the fine-grained quantization, we adopt per-token (Yao et al., 2022) and per-group (Yao et al., 2023) quantization.

*Notation:* We use INT8, INT6, INT4 to denote the bitwidth of activation and weight. Specifically, INT8* refers to per-tensor weight quantization. And per-token and per-group quantization will be marked in the table below.

**Models and tasks.** We conduct experiments on both small and large language models. First, BERT models (base and large versions) are evaluated on the GLUE benchmark (Wang et al., 2018a). Second, four of the largest OPTs ranging from 13B to 175B, biggest BLOOM (Scao et al., 2022) and BLOOMZ (Muennighoff et al., 2022) boasting 176 billion parameters, and LLaMA (Touvron et al., 2023) models including 7B, 13B, 30B, 65B sizes are chosen as representatives. Zero-shot tasks including language modeling, multiple choice, commonsense reasoning, etc. are selected for evaluation. The evaluation code is based on `lm-harness-evaluation`[1].

**Baselines.** For BERT, we adopt classical PTQ techniques as baselines, including MinMax, Per-

centile (Wu et al., 2020), OMSE (Choukroun et al., 2019), and recent works on BERT quantization including PEG (Bondarenko et al., 2021), and Outlier Suppresion (Wei et al., 2022b). For large models including OPT, BLOOM, and LLaMA, we mainly compare with recent works including Zero-Quant (Yao et al., 2022), and SmoothQuant (Xiao et al., 2022). For details, readers can refer to Appendix C.

**Implementation.** We randomly select 128 samples from the training dataset, in-domain data for the GLUE benchmark, and PILE (Gao et al., 2020) dataset for zero-shot tasks. A batch of data is first used to calculate effective shifting and scaling vectors. Then, calibration is conducted. More details can be found in Appendix C.

### 5.2 Standard quantization with OS+

In this section, we show how OS+ can help standard quantization achieve satisfying results from both the small models and LLMs aspects.

| Method | CoLA | MNLI | QNLI | SST-2 | STS-B | Avg. |
|---|---|---|---|---|---|---|
| **FP32** | 59.6 | 84.9 | 91.8 | 93.4 | 89.5 | 83.8 |
| **INT8*** | | | | | | |
| MinMax | 52.3 | 81.3 | 89.0 | 91.1 | 86.2 | 79.5 |
| OMSE | 54.8 | 82.1 | 89.7 | 91.3 | 87.7 | 81.6 |
| PEG | 59.4 | 81.3 | 91.1 | 92.7 | 87.9 | 82.5 |
| OS | 60.3 | 83.9 | 90.2 | **92.9** | 88.2 | 83.0 |
| OS+ | **60.9** | **84.4** | **91.1** | 92.7 | **88.3** | **83.5** |
| **INT6** | | | | | | |
| OMSE | 35.4 | 73.7 | 84.7 | 86.3 | 85.8 | 73.5 |
| Percentile | 37.3 | 72.1 | 79.4 | 87.3 | 86.8 | 72.9 |
| OS | 54.4 | 81.8 | 89.8 | 91.9 | 88.7 | 81.2 |
| OS+ | **56.0** | **84.5** | **90.9** | **92.4** | **89.5** | **82.8** |
| **INT4** | | | | | | |
| OMSE | 4.7 | 38.5 | 52.2 | 50.3 | 0.2 | 41.1 |
| Percentile | 7.0 | 53.0 | 61.5 | 77.1 | 66.1 | 57.0 |
| OS | 28.5 | 57.9 | 72.5 | 80.4 | 67.8 | 62.7 |
| OS+ | **50.0** | **80.2** | **85.4** | **91.4** | **86.5** | **78.2** |

Table 1: PTQ performance of BERT-base models. MNLI and STS-B report the combined score. **Avg.** indicates the averaged results of 8 tasks on GLUE benchmark (details in Appendix B). ∗ means per-tensor quantization for weight. OS indicates Outlier Suppression for short.

**BERT.** Table 1 gives prediction results of common PTQ algorithms. Most methods perform well on INT8* but fail on lower bits while our approach consistently achieves superior outcomes. Compared to Wei et al. (2022b), our method outperforms by 1.6% and 15.5% on 6-bit and 4-bit, respectively. In summary, our approach can achieve near-floating point performance on high bits and

[1] https://github.com/EleutherAI/lm-evaluation-harness

| Model | Method | PIQA (↑) | | | Winogrande (↑) | | | HellaSwag (↑) | | | LAMBADA (↑) | | |
|---|---|---|---|---|---|---|---|---|---|---|---|---|---|
| | | FP16 | INT8* | INT6 | FP16 | INT8* | INT6 | FP16 | INT8* | INT6 | FP16 | INT8* | INT6 |
| OPT-13B | ZeroQuant | | 54.1 | 53.0 | | 52.1 | 51.1 | | 26.5 | 25.8 | | 42.9 | 0.0 |
| | SmoothQuant | 75.8 | 76.0 | 73.5 | 65.1 | 64.9 | 60.3 | 52.5 | 52.2 | 49.2 | 68.6 | 68.3 | 65.2 |
| | OS+ | | **76.4** | **75.8** | | **65.0** | **64.0** | | **52.3** | **51.7** | | **68.3** | **65.7** |
| OPT-30B | ZeroQuant | | 54.2 | 52.0 | | 51.8 | 51.8 | | 26.4 | 25.7 | | 9.7 | 0.0 |
| | SmoothQuant | 77.6 | 77.2 | 66.7 | 68.5 | **68.2** | 55.0 | 54.3 | 54.2 | 37.4 | 71.5 | **71.0** | 13.4 |
| | OS+ | | **77.4** | **77.4** | | 68.0 | **68.9** | | **54.2** | **53.7** | | 70.8 | **69.6** |
| OPT-66B | ZeroQuant | | 53.2 | 51.9 | | 50.7 | 48.0 | | 26.1 | 25.7 | | 0.6 | 0.0 |
| | SmoothQuant | 78.7 | 78.3 | 52.0 | 68.9 | 68.3 | 52.1 | 56.4 | 55.9 | 26.5 | 73.9 | 72.9 | 0.0 |
| | OS+ | | **78.7** | **77.5** | | **69.0** | **69.4** | | **56.2** | **55.8** | | **73.0** | **72.7** |
| OPT-175B | ZeroQuant | | 52.3 | 53.1 | | 50.2 | 49.1 | | 25.4 | 25.6 | | 0.0 | 0.0 |
| | SmoothQuant | 79.7 | **79.7** | 52.6 | 72.5 | 71.2 | 49.1 | 59.3 | 58.9 | 26.0 | 74.7 | 74.6 | 0.5 |
| | OS+ | | 79.6 | **80.0** | | **72.5** | **71.7** | | **59.2** | **58.5** | | **74.7** | **74.2** |
| BLOOM-176B | ZeroQuant | | 76.0 | 61.2 | | 69.4 | 52.0 | | 54.8 | 30.5 | | 67.8 | 7.5 |
| | SmoothQuant | 78.8 | 77.7 | 76.7 | 70.3 | 68.6 | 67.6 | 55.9 | 54.1 | 52.1 | 67.7 | **69.2** | 60.2 |
| | OS+ | | **78.4** | **78.1** | | **69.8** | **70.3** | | **55.2** | **54.8** | | 68.0 | **69.2** |
| BLOOMZ-176B | ZeroQuant | | 79.1 | 54.0 | | 70.9 | 49.6 | | 56.3 | 28.2 | | 67.6 | 1.4 |
| | SmoothQuant | 80.6 | 79.7 | **80.0** | 72.5 | 70.8 | 69.9 | 57.1 | 56.3 | 55.0 | 67.8 | 68.7 | 65.2 |
| | OS+ | | **79.9** | 79.9 | | **71.3** | **70.6** | | **56.7** | **56.4** | | **68.8** | **69.2** |

Table 2: Comparison among different techniques in terms of accuracy on four zero-shot tasks. INT8* specifically means per-tensor quantization for weights compared to INT8. More tasks are put in Appendix B due to space limit.

reduce the performance gap to 5.6% on 4-bit.

**OPT and BLOOM.** With standard quantization, we list 8-bit and 6-bit accuracy in Table 2. It can be observed that OS+ outperforms ZeroQuant by a large margin. While SmoothQuant suffers from non-negligible accuracy drops on much harder settings like the 6-bit 175B model with significantly severe outliers, ours still gives enjoyable results, owning 32.5% upswings on HellaSwag task, 27.4% boost on PIQA. Results of BLOOM models indicate that their quantization challenges are less severe than OPTs with smaller accuracy drops across methods. Our approach still beats the best of others by about 2% points on 6-bit. To conclude, with standard quantization, ours is indeed close to FP results on 8-bit and exhibits around 1 point accuracy degradation on 6-bit.

### 5.3 Fine-grained quantization with OS+

Here, OS+ is combined with fine-grained quantization to validate its wide application and go extremely low bit setting like 4-bit quantization.

**Per-token Quantization.** Per-token quantization (Yao et al., 2022), which customizes quantization parameters for individual tokens, can bring better predictions, especially for lower-bit quantization and longer output like WikiText2 (Merity et al., 2017). We opt for LLaMA models for validation. It's worth noting that the structure of LLaMA differs from others in its design of element-wise multiplication of two activations as the input to the final layer in FFN, potentially resulting in very

large signals, even exceeding 600. Given such a challenge, we provide experiments both with and without quantization of this layer in Table 3 and Table 10, respectively. In both tables, we highlight our lossless performance on 6-bit quantization while SmoothQuant still suffers in Table 10. Also, it shows the superior performance of OS+ on 4-bit (e.g., 10.58% improvement on Winogrande, 10.04 PPL decrease on WikiText2).

**Per-group Quantization.** Additionally, per-group quantization (Yao et al., 2023), which tailors quantization parameters for each group of elements, is a more fine-grained way. Recognizing the difficulties of 4-bit quantization for OPTs, we illustrate an example by adopting per-group quantization with relatively large group sizes of 1024 and 512. Fig. 3 shows that OS+ continues to outperform other methods and can be more competitive under harder cases such as a group size of 1024.

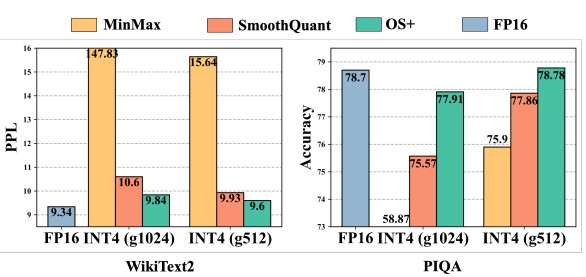

Figure 3: Results of 4-bit quantization with group size set to 1024 and 512, respectively.

| Model | Method | PIQA (↑) | | | Winogrande (↑) | | | HellaSwag (↑) | | | WikiText2 (↓) | | |
|---|---|---|---|---|---|---|---|---|---|---|---|---|---|
| | | FP16 | INT6 | INT4 | FP16 | INT6 | INT4 | FP16 | INT6 | INT4 | FP16 | INT6 | INT4 |
| LLaMA-1-7B | MinMax | | 77.26 | 55.98 | | 66.54 | 49.64 | | 71.78 | 32.28 | | 6.00 | 473.97 |
| | SmoothQuant | 77.37 | 77.18 | 70.08 | 66.93 | 65.51 | 52.96 | 72.99 | 72.10 | 58.13 | 5.68 | 5.85 | 16.87 |
| | OS+ | | **77.48** | **72.31** | | **67.01** | **56.67** | | **72.32** | **61.24** | | **5.76** | **14.17** |
| LLaMA-1-13B | MinMax | | 78.56 | 50.65 | | 69.53 | 50.28 | | 75.26 | 26.34 | | 5.58 | 3410.45 |
| | SmoothQuant | 79.05 | 78.45 | 66.49 | 70.09 | **69.69** | 51.78 | 76.22 | 75.20 | 58.95 | 5.09 | 5.25 | 56.75 |
| | OS+ | | **78.73** | **75.03** | | 69.53 | **61.17** | | **75.74** | **67.21** | | **5.22** | **18.95** |
| LLaMA-1-30B | MinMax | | 78.40 | 50.00 | | 72.45 | 50.12 | | 77.25 | 27.09 | | 5.09 | 2959.15 |
| | SmoothQuant | 80.09 | 78.78 | 71.55 | 72.77 | 73.01 | 54.54 | 79.21 | 78.13 | 60.97 | 4.10 | 4.40 | 51.47 |
| | OS+ | | **79.98** | **73.01** | | **73.64** | **60.38** | | **78.77** | **68.03** | | **4.30** | **22.61** |
| LLaMA-1-65B | MinMax | | 77.58 | 50.27 | | 69.46 | 49.33 | | 78.72 | 24.59 | | 5.25 | 14584.66 |
| | SmoothQuant | 80.85 | 78.40 | 65.02 | 77.11 | 74.30 | 51.14 | 80.73 | 78.57 | 59.78 | 3.56 | 3.77 | 19.37 |
| | OS+ | | **80.47** | **74.43** | | **75.14** | **61.72** | | **79.76** | **67.65** | | **3.65** | **9.33** |

Table 3: Comparison in terms of normalized accuracy, accuracy, normalized accuracy and perplexity (PPL), respectively (Touvron et al., 2023). Techniques are equipped with **per-token quantization** (Yao et al., 2022). More results are put in Appendix B.

## 5.4 Ablation study

**Design choices of scaling values.** In this section, we compare different scaling vector designs. In Table 4, the second row displays results without attention post-processing Eq. (7). Summing the losses of multiple linear layers, as shown, proves unwise, resulting in performance declines of about 2% and 10% on OPTs. The third row removes the outlier threshold and instead learns scaling values directly. We find this process is unstable and requires suitable hyperparameters, causing failure on LLMs. As mentioned in Sec. 4.2.2, This instability may stem from suboptimal scaling values for normal channels with varying magnitudes.

**Effect of each operation.** From Table 5, it can be observed clearly that by removing the shifting operation, the accuracy drops by about 1%-3% under difficult settings. This is because, without channel-wise shifting that initially smooths the quantization challenge, scaling factors struggle to suppress outliers effectively while producing the tolerable weight quantization burden. Furthermore, when excluding scaling effects, performance decreases significantly, with even crashed results on LLMs.

## 5.5 Analysis

**Different activation scaling.** Because scaling values act in both the activation and weights, reducing quantization error for individual tensors can not guarantee the minimum output change, which encapsulates their information to later forward pass. For example, in Table 6, Outlier Suppression with fixed scaling values has the smallest quantization error for weight. SmoothQuant with a heuristic way has the smallest quantization error for activation. However, both of them did not bring the smallest quantization error for the output. This reveals the importance of directly optimizing according to the output, which is what our method exactly does. Thus, we can enjoy the best final performance.

**Model storage and accuracy.** Inspired by a variety of models with diverse sizes, we also study the relationship between their storage and accuracy under quantization settings. Focusing on one kind of model with distinct quantization bit-width, Fig. 4 shows that 8-bit quantization which cuts storage by about half, can generally maintain original performance, and 6-bit quantization can lead to less performance drop on larger models. Moreover, considering fixed storage constraints, we discover that quantized big models typically outperform small FP models. These observations can relate to model robustness, which implies that large models can

| Method | OPT-66B (INT6) | | BERT (INT4) | |
|---|---|---|---|---|
| | PIQA | Winogrande | SST-2 | MNLI |
| scaling | **76.5** | **66.5** | **89.3** | **77.7** |
| - attention post process | 74.5 | 57.4 | 89.1 | 77.1 |
| - outlier threshold | Fail | Fail | 83.2 | 65.2 |

Table 4: Design choices of scaling factor. The second row removes the attention post process in optimization objective. The third row chooses to learn the scaling vector directly rather than alternately optimize the outlier threshold.

| Method | OPT-66B (INT6) | | BERT (INT4) | |
|---|---|---|---|---|
| | PIQA | Winogrande | SST-2 | MNLI |
| **Ours** | **77.5** | **69.4** | **91.4** | **80.2** |
| - shifting | 76.5 | 66.5 | 89.3 | 77.7 |
| - shifting - scaling | 54.7 | 49.4 | 82.3 | 63.7 |

Table 5: Effect of scaling and shifting operations.

| Method | activation | | weight | | Output change |
| --- | --- | --- | --- | --- | --- |
| | range | MSE | range | MSE | MSE |
| original | (-93.9, 31.6) | 209.8 | (-0.13, 0.13) | 0.001 | 18061.5 |
| OS | (-23.5,15.7) | 142.9 | (-0.40, 0.41) | **0.006** | 6182.52 |
| SQ | (-3.5, 2.0) | **3.65** | (3.4, 3.5) | 0.43 | 3535.86 |
| Our scaling | (-8.4, 8.4) | 48.54 | (1.2, 1.3) | 0.02 | **1334.89** |

Table 6: Detailed analysis of different techniques from the activation scaling aspect. OS indicates Outlier Suppression and SQ indicates SmoothQuant.

benefit from compression more if special outliers are handled well.

**Computation Complexity.** We explain our computation complexity of calibration and deployment phases. For the calibration process, OS+ is efficient, and able to generate scaling and shifting values in about 20 minutes for OPT-175B offline. Moreover, due to the equivalent transformation, our method does not demand additional training and can be applied in a post-training setting. For deployment, we discuss inference efficiency with latency performance evaluated using (NVIDIA, 2022). As mentioned before, our channel-wise shifting and scaling can be implemented by updating previous parameters, and be migrated to subsequent weights. For LLMs, our transformation does not introduce any extra computation burden and leads to favorable latency improvements, as demonstrated in a 1.5× speedup in Fig. 5. Only BERT models additionally replace the identity function in the residual connection with channel-wise multiplication and addition. Such overhead is minimal, as shown in Fig. 5, resulting in comparable latency speedup.

## 6 Conclusion

We present the Outlier Suppression+ framework for addressing asymmetric and consistent outliers

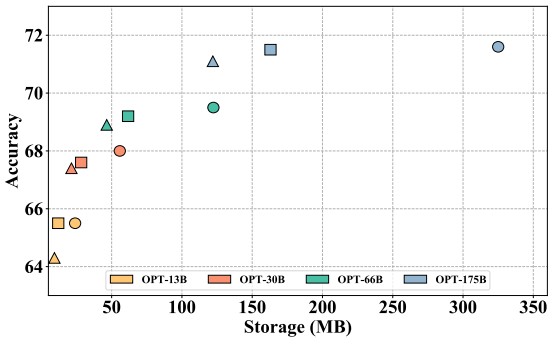

Figure 4: Averaged accuracy on PIQA, Winogrande, LAMBADA, and HellaSwag of OPTs with different storages. We draw circles, rectangles, and triangles to refer to FP16, the 8-bit and 6-bit models with quantized activation and weight.

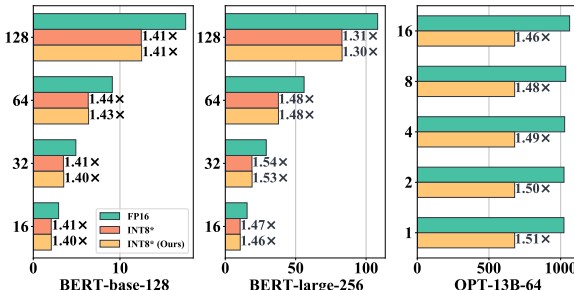

Figure 5: Real latency (x-axis) of our transformed 8-bit models, 8-bit and FP16 original models over different batch sizes (y-axis). BERT-large-256 refers to the BERT-large model with sequence length set to 256 while for OPT-13B-64, 64 means output length with input length set to 512. Bold numbers indicate quantization speedup.

in LLMs and other transformers. Our framework is simple to use, consisting of both scaling and shifting operations, which can be efficiently and effectively implemented. Experiments demonstrate the efficacy of our methods for suppressing outliers.

## Limitations

While we have observed features of outliers and devised methods to deal with them, the underlying reasons for their emergence and attributes have not been fully understood. This may require an in-depth analysis of the training pipeline, including the procedure and hyperparameters. Such investigations are time-consuming but can benefit both FP and quantized scenarios.

## Ethics Statement

Our Outlier Suppression+ framework aims to improve the quantization performance of transformer language models. It can boost the development of practical and green machine learning and does not incur extra ethical concerns.

## Acknowledgment

We sincerely thank the anonymous reviewers for their sincere reviews and valuable suggestions to make this better. We also thank Qi Zhang for the insightful discussion and Jing Liu for helping to build the code of LLaMA. This work was supported by the National Natural Science Foundation of China (No. 62022009), National Natural Science Foundation of China (No. 62306025), the State Key Laboratory of Software Development Environment (SKLSDE-2022ZX-23).

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

# A  Related work

**Quantization.** Compression has become more and more popular these days (Han et al., 2015; Hinton et al., 2015; Hu et al., 2021; Liu et al., 2022; Xu et al., 2023; Chen et al., 2023). One of its effective techniques called quantization (Jacob et al., 2018) employs low-bit representations for activation and weight in neural networks. Researchers categorize this approach into two pipelines: post-training quantization (PTQ) and quantization-aware training (QAT). QAT (Courbariaux et al., 2015; Choi et al., 2018; Esser et al., 2019; Li et al., 2019; Gong et al., 2019; Shen et al., 2021; Zhang et al., 2018) trains the quantized model end-to-end, necessitating significant GPU resources and the entire training dataset. In contrast, PTQ (Choukroun et al., 2019; Wu et al., 2020; Banner et al., 2018; Wang et al., 2020; Zhao et al., 2019; Nagel et al., 2019) only requires hundreds of samples and limited resource consumption, producing a calibrated model quickly. Recently, several works (Nagel et al., 2020; Hubara et al., 2021; Li et al., 2021; Wei et al., 2022a) proposed to adjust models slightly for improved PTQ performance. Besides, other types of quantization include zero-shot quantization without real calibration data (Cai et al., 2020; Zhang et al., 2021; Guo et al., 2022), mixed-precision with mixed bit-width (Dong et al., 2019; Cai and Vasconcelos, 2020), and FP8 data type (Wang et al., 2018b; Kuzmin et al., 2022; Micikevicius et al., 2022; Jin et al., 2022).

**Quantization of transformer language models.** Recently, there has been a growing interest in the quantization of transformer language models. In the context of QAT, Zafrir et al. (2019) first explores 8-bit quantization for BERT-like models. Shen et al. (2020) introduces group-wise quantization and studies mixed-precision quantization based on Hessian information. Bai et al. (2020); Zhang et al. (2020); Qin et al. (2022) combine distillation strategies with quantization. Kim et al. (2021) approximates the nonlinear function in transformer architectures to enable integer-only inference. Fan et al. (2020) incorporates quantization noise for enhancement. Additionally, Tao et al. (2022) investigates the challenges of quantizing generative models.

In the realm of PTQ, researchers have discovered that the poor performance of these models should be attributed to extreme outliers in activations. These outliers exhibit special characteristics from both channel and token aspects. *In terms of channels*, outliers consistently emerge in certain channels over different inputs. Bondarenko et al. (2021) employs a per-embedding-group quantization scheme that uses different quantization parameters for distinct channel groups, while Dettmers et al. (2022) suggests utilizing FP16 representations for problematic channels holding signals over 6. Wei et al. (2022b) identifies this feature lying in LayerNorm's output and migrates the scaling parameter of LayerNorm to subsequent modules to attenuate outliers. Xiao et al. (2022) proposes calculating scaling values by equalizing ranges between activations and weights and evaluates on large language models. Guo et al. (2023) discards normal values adjacent to outliers, making room for outliers with customized GPU support. Compared to them, we design the scaling factors that concern the interactive results of troublesome activation and following weights to scale down channels with outliers offline. Also, we notice the asymmetric presentation of outliers and design a shifting operation. While we operate on corresponding channels between weights and activation, a later work (Liu et al., 2023) adopts the splitting and merging operations to transfer the quantization burden of outlier channels to opposite channels of weights, which might encourage us to design a more flexible technique without the same or opposite channel index requirement. *In terms of tokens*, different tokens exhibit varying degrees of outliers. Dettmers et al. (2022); Yao et al. (2022) introduce a novel scheme called per-token quantization that dynamically computes quantization parameters for each token. Wei et al. (2022b) investigates the clipping impact of outliers and recommends finding an appropriate clipping range in a token-wise manner.

Besides, some studies focus on weight quantization, such as Dettmers and Zettlemoyer (2022); Frantar et al. (2022); Zeng et al. (2022); Lin et al. (2023) and some including Yuan et al. (2021); Li et al. (2022), investigate the quantization of Vision Transformer (ViT) models. Interestingly, several studies (Kovaleva et al., 2021; Puccetti et al., 2022) explore the underlying reasons for emerging outliers and trace them back to the pre-training phase.

# B  Supplementary experiments

**BERT-base.** We provide detailed results of BERT-base models on GLUE benchmarks in Table 7. Interestingly, we find that models which are sensitive

**Algorithm 1:** Outlier Suppression+

**Input:** Problematic output $\boldsymbol{X}$ of LayerNorm with parameters $\boldsymbol{\gamma}, \boldsymbol{\beta}$, subsequent module $M$ with weight $\boldsymbol{W}$ and bias $\boldsymbol{b}$, grid search iteration $K$.

{1. Effective shifting and scaling:}

$\boldsymbol{z} = \frac{\min(\boldsymbol{X}_{:,j}) + \max(\boldsymbol{X}_{:,j})}{2}$      $\triangleright$ Effective shifting vector.

$loss^* = \text{INF}$

**for** $k = 1$ *to* $K$ **do**

     $t = \max(\boldsymbol{X} - \boldsymbol{z}) \cdot \frac{k}{K}$,      $\triangleright$ Enumerate outlier threshold.

     $\boldsymbol{s}_j = \max(1.0, \frac{\max(\boldsymbol{X}_{:,j} - \boldsymbol{z}_j)}{t})$

     Calculate $loss_k$ based on Eq. (6), Eq. (7).

     **if** $loss^* > loss_k$ **then**

         $loss^* = loss_k, \boldsymbol{s}^* = \boldsymbol{s}$      $\triangleright$ Effective scaling factors.

{2. Equivalent shifting and scaling:}

$\widetilde{\boldsymbol{\beta}} = (\boldsymbol{\beta} - \boldsymbol{z}) \oslash \boldsymbol{s}^*, \widetilde{\boldsymbol{\gamma}} = \boldsymbol{\gamma} \oslash \boldsymbol{s}^*$      $\triangleright$ Fuse $\boldsymbol{z}, \boldsymbol{s}^*$ into former operations.

$\widetilde{\boldsymbol{b}} = \boldsymbol{z}\boldsymbol{W}^\top + \boldsymbol{b}, \widetilde{\boldsymbol{W}} = \boldsymbol{W} \odot \boldsymbol{s}^*$      $\triangleright$ Update following modules.

**return** Transformed LayerNorm and subsequent module;

| Method | CoLA (Matt.) | MNLI (acc m/mm) | MRPC (f1/acc) | QNLI (acc) | QQP (f1/acc) | RTE (acc) | SST-2 (acc) | STS-B (Pear./Spear.) | Avg. |
|---|---|---|---|---|---|---|---|---|---|
| **FP32** | 59.6 | 84.9/84.8 | 91.4/87.8 | 91.8 | 87.8/90.9 | 72.6 | 93.4 | 89.7/89.3 | 83.8 |
| **INT8*** | | | | | | | | | |
| MinMax | 52.3 | 80.9/81.7 | 85.3/80.9 | 89.0 | 84.8/88.6 | 68.2 | 91.1 | 84.7/87.6 | 79.5 |
| OMSE | 54.8 | 81.9/82.2 | 89.7/86.0 | 89.7 | 86.1/89.5 | 72.2 | 91.3 | 87.2/88.2 | 81.6 |
| PEG | 59.4 | 81.3 | 88.5 | 91.1 | **89.4** | 69.3 | 92.7 | 87.9 | 82.5 |
| OS | 60.3 | 83.8/84.0 | 90.4/87.0 | 90.2 | 87.3/90.4 | 71.1 | **92.9** | 87.8/88.7 | 83.0 |
| **Ours** | **60.9** | **84.4/84.4** | **90.6/87.2** | **91.1** | 87.1/90.6 | **73.3** | 92.7 | **87.7/88.9** | **83.5** |
| **INT8** | | | | | | | | | |
| MinMax | 57.1 | 82.8/83.5 | 89.9/85.8 | 90.8 | 87.8/90.7 | 69.7 | 92.8 | 86.8/88.6 | 82.3 |
| OMSE | 57.2 | 84.0/84.3 | 90.1/85.8 | 91.1 | 87.6/90.5 | 72.2 | 92.2 | 87.9/88.7 | 82.9 |
| Percentile | 57.1 | 83.9/84.1 | 90.7/86.7 | 91.3 | 87.7/90.7 | 71.1 | 93.4 | 87.7/88.7 | 82.9 |
| OS | **61.6** | 84.4/84.5 | **91.4/87.8** | 91.5 | **87.9/90.8** | **72.2** | 93.8 | 89.2/89.0 | **84.0** |
| **Ours** | 60.3 | **84.8/84.5** | 90.5/87.0 | **91.6** | 87.5/90.8 | 71.5 | 93.6 | **89.3/89.2** | 83.6 |
| **INT6** | | | | | | | | | |
| MinMax | 17.7 | 32.5/32.5 | 0.7/31.9 | 65.2 | 40.9/69.0 | 48.0 | 82.0 | 59.8/60.3 | 47.1 |
| OMSE | 35.4 | 74.0/73.3 | 81.5/76.5 | 84.7 | 76.1/82.1 | 64.3 | 86.3 | 85.6/86.1 | 73.5 |
| Percentile | 37.3 | 72.4/71.7 | 85.1/79.9 | 79.4 | 72.6/80.2 | 61.7 | 87.3 | 86.4/87.3 | 72.9 |
| OS | 54.4 | 82.0/81.7 | 87.5/83.3 | 89.8 | 84.7/88.9 | 70.8 | 91.9 | 88.7/88.6 | 81.2 |
| **Ours** | **56.0** | **84.6/84.4** | **90.0/86.3** | **90.9** | **87.0/90.5** | **71.8** | **92.4** | **89.6/89.4** | **82.8** |
| **INT4** | | | | | | | | | |
| MinMax | -6.6 | 32.6/32.7 | 0.0/31.6 | 50.6 | 53.8/36.8 | 47.7 | 50.9 | -0.5/-0.5 | 29.5 |
| OMSE | 4.7 | 38.5/38.4 | 81.3/69.1 | 52.2 | 45.2/50.9 | 59.9 | 50.3 | 0.1/-0.4 | 41.1 |
| Percentile | 7.0 | 52.6/53.5 | 83.0/75.7 | 61.5 | 44.7/68.3 | 55.6 | 77.1 | 65.9/66.3 | 57.0 |
| OS | 28.5 | 57.5/58.3 | 83.9/75.7 | 72.5 | 45.4/70.8 | 56.7 | 80.4 | 67.8/67.9 | 62.7 |
| **Ours** | **50.0** | **80.6/79.9** | **87.6/83.1** | **85.4** | **85.0/77.5** | **65.7** | **91.4** | **86.4/86.5** | **78.2** |

Table 7: PTQ performance of BERT-base models on GLUE benchmark. ∗ means per-tensor quantization for weight. OS indicates Outlier Suppression for short.

| Method | CoLA (Matt.) | MNLI (acc m/mm) | MRPC (f1/acc) | QNLI (acc) | QQP (f1/acc) | RTE (acc) | SST-2 (acc) | STS-B (Pear./Spear.) | Avg. |
|---|---|---|---|---|---|---|---|---|---|
| FP32 | 63.3 | 86.7/85.9 | 91.6/88.0 | 92.2 | 88.1/91.1 | 74.0 | 93.5 | 90.3/90.1 | 84.9 |
| **INT8*** | | | | | | | | | |
| MinMax | 62.4 | 72.0/73.0 | 76.3/72.8 | 87.0 | 66.5/80.4 | 46.9 | 92.2 | 58.6/52.1 | 71.5 |
| OMSE | 59.9 | 82.7/83.5 | 87.8/83.8 | 89.0 | 79.2/86.2 | 47.3 | 92.0 | 83.9/83.3 | 78.1 |
| Percentile | 61.3 | 84.5/84.0 | 91.6/88.9 | 91.6 | 85.9/89.4 | 69.3 | 92.4 | 88.3/88.1 | 83.1 |
| OS | **62.3** | 85.1/84.5 | 90.1/86.0 | 91.1 | 87.0/90.3 | **75.1** | 92.4 | 88.7/88.4 | 83.9 |
| **Ours** | 62.2 | **85.9/85.2** | 90.9/87.0 | 92.2 | 87.8/90.8 | 71.8 | **93.3** | 89.3/89.3 | **84.1** |
| **INT6** | | | | | | | | | |
| MinMax | 5.6 | 32.0/32.0 | 50.2/46.1 | 50.2 | 0.0/63.2 | 49.5 | 53.0 | 5.0/4.8 | 38.1 |
| OMSE | 14.0 | 59.3/58.4 | 86.1/78.7 | 79.5 | 52.5/73.5 | 54.9 | 74.8 | 44.0/37.9 | 59.8 |
| Percentile | 16.4 | 63.5/63.8 | 82.0/77.2 | 87.0 | 44.8/70.7 | 49.8 | 81.7 | 65.7/67.8 | 62.8 |
| OS | 24.1 | 71.3/71.7 | 85.5/79.4 | 80.8 | 68.8/78.3 | 47.3 | 82.3 | 61.1/62.0 | 65.4 |
| **Ours** | **60.9** | **86.3/85.4** | **91.8/88.2** | **92.0** | **87.7/90.8** | **71.5** | **93.7** | **86.7/85.6** | **83.7** |

Table 8: PTQ performance of BERT-large models on GLUE benchmark. ∗ means per-tensor quantization for weight. OS indicates Outlier Suppression for short.

to different learning hyperparameters during the fine-tuning phase, such as CoLA and RTE, also exhibit less favorable quantization outcomes. This suggests a possible relationship between quantization and robustness.

**BERT-large.** We also conduct experiments on BERT-large models in Table 8. Results across methods indicate that quantizing BERT-large models is more challenging (e.g., MinMax suffers from a considerable accuracy drop (about 13%) on INT8* compared to BERT-base, and Outlier Suppression also fails on the 6-bit setting). Fortunately, with Outlier Suppression+, the results can be improved, yielding an 18.7% enhancement.

**OPT.** Here, we provide results of OPTs on more tasks. Table 9 is the supplement for Table 2, which further shows consistent performance enhancement of OS+.

**LLaMA.** Recall that we conduct experiments on LLaMA with two different settings in the fine-grained quantization section. Table 10 gives the results when quantizing the special and challenging structure (the last layer of FFN) in LLaMA models. It can be observed that ours still earns near-floating-point performance on 6-bit quantization and beats others by about 5%∼14% in terms of averaged accuracy of the first four tasks, and even four times PPL decrease for WikiText2. By comparing with the easier setting Table 3, we find that the special structure with large signals really leads to much lower 4-bit outcomes across methods, especially

for MinMax and SmoothQuant, which makes us think of model design, training techniques, and efficient fine-tuning for quantization.

## C  Implementation details

### C.1  OS+

In this section, we provide detailed descriptions of our implementation with the core part distilled in algorithm 1.

**BERT.** On the GLUE benchmark, fine-tuned FP models are used for quantization. We randomly select 128 samples and set the batch size to 32. First, a batch of data is used to calculate the effective shifting and scaling signals for problematic activations, especially outputs after LayerNorm here. Then shifting and scaling vectors are fused into former operations and absorbed in later modules. On fused models, we apply the calibration procedure. Particularly, on BERT models, due to the great variance of token range as discussed in Yao et al. (2022); Wei et al. (2022b), we incorporate the Token-Wise Clipping proposed in Outlier Suppression which is an orthogonal technique and weakens outliers from the token aspect.

**OPTs.** For OPTs, we quantize pre-trained models and evaluate them on zero-shot tasks. 128 samples are randomly extracted from one of the train datasets, namely the PILE dataset. As we have observed that LayerNorm produces severe asymmetric outliers on certain channels, the pro-

| Name | Method | OPT-13B | | | | OPT-30B | | | | OPT-66B | | | | OPT-175B | | | |
|---|---|---|---|---|---|---|---|---|---|---|---|---|---|---|---|---|---|
| | | FP16 | INT8* | INT8 | INT6 | FP16 | INT8* | INT8 | INT6 | FP16 | INT8* | INT8 | INT6 | FP16 | INT8* | INT8 | INT6 |
| PIQA | LLM.int8()♣ | | - | 75.8 | - | | - | 77.3 | - | | - | 78.7 | - | | - | 79.6 | - |
| | ZeroQuant♣ | 75.8 | 54.1 | - | 53.0 | 77.6 | 54.2 | - | 52.0 | 78.7 | 53.2 | - | 51.9 | 79.7 | 52.3 | - | 53.1 |
| | SmoothQuant | | 76.0 | - | 73.5 | | 77.2 | - | 66.7 | | 78.3 | - | 52.0 | | **79.7** | - | 52.6 |
| | Ours | | **76.4** | 75.9 | **75.8** | | **77.4** | 77.6 | **77.4** | | **78.7** | 78.6 | **77.5** | | 79.6 | 79.5 | **80.0** |
| LAMBADA | LLM.int8()♣ | | - | 68.4 | - | | - | 71.4 | - | | - | 73.8 | - | | - | 74.6 | - |
| | ZeroQuant♣ | 68.6 | 0.0 | - | 0.0 | 71.5 | 0.0 | - | 0.0 | 73.9 | 0.0 | - | 0.0 | 74.7 | 0.0 | - | 0.0 |
| | SmoothQuant | | 68.3 | - | 65.2 | | **71.0** | - | 13.4 | | 72.9 | - | 0.0 | | **74.6** | - | 0.5 |
| | Ours | | **68.3** | 68.4 | **65.7** | | 70.8 | 70.8 | **69.6** | | **73.0** | 73.4 | **72.7** | | 74.7 | 74.5 | **74.2** |
| HellaSwag | LLM.int8()♣ | | - | 52.4 | - | | - | 54.3 | - | | - | 56.3 | - | | - | 59.2 | - |
| | ZeroQuant♣ | 52.5 | 26.5 | - | 25.8 | 54.3 | 26.4 | - | 25.7 | 56.4 | 26.1 | - | 25.7 | 59.3 | 25.4 | - | 25.6 |
| | SmoothQuant | | 52.2 | - | 49.2 | | 54.2 | - | 37.4 | | 55.9 | - | 26.5 | | 58.9 | - | 26.0 |
| | Ours | | **52.3** | 52.5 | **51.7** | | **54.2** | 54.2 | **53.7** | | **56.2** | 56.3 | **55.8** | | **59.2** | 59.3 | **58.5** |
| Winogrande | LLM.int8()♣ | | - | 64.8 | - | | - | 68.1 | - | | - | 68.5 | - | | - | 72.3 | - |
| | ZeroQuant♣ | 65.1 | 52.1 | - | 51.1 | 68.5 | 51.8 | - | 51.8 | 68.9 | 50.7 | - | 48.0 | 72.5 | 50.2 | - | 49.1 |
| | SmoothQuant | | 64.9 | - | 60.3 | | **68.2** | - | 55.0 | | 68.3 | - | 52.1 | | 71.2 | - | 49.1 |
| | Ours | | **65.0** | 65.3 | **64.0** | | 68.0 | 68.5 | **68.9** | | **69.0** | 68.8 | **69.4** | | **72.5** | 72.5 | **71.7** |
| ARC (Challenge) | LLM.int8()♣ | | - | 33.5 | - | | - | 34.7 | - | | - | 37.0 | - | | - | 40.9 | - |
| | ZeroQuant♣ | 32.8 | 19.3 | - | 20.7 | 34.6 | 19.8 | - | 20.6 | 37.3 | 20.8 | - | 20.4 | 40.3 | 21.8 | - | 20.6 |
| | SmoothQuant | | 32.1 | - | 30.6 | | 33.8 | - | 26.7 | | 36.5 | - | 21.9 | | **40.5** | - | 21.2 |
| | Ours | | **33.5** | 33.3 | **32.7** | | **34.5** | 34.7 | **34.6** | | **37.5** | 37.2 | **37.0** | | 40.3 | 39.9 | **41.0** |
| ARC (Easy) | LLM.int8()♣ | | - | 67.3 | - | | - | 69.7 | - | | - | 71.8 | - | | - | 74.8 | - |
| | ZeroQuant♣ | 67.3 | 27.5 | - | 25.0 | 70.1 | 30.5 | - | 25.0 | 71.7 | 29.7 | - | 26.0 | 74.9 | 24.0 | - | 25.6 |
| | SmoothQuant | | 66.2 | - | 62.2 | | 69.7 | - | 55.8 | | 70.5 | - | 27.8 | | 74.1 | - | 28.8 |
| | Ours | | **67.3** | 66.8 | **67.0** | | **70.1** | 70.0 | **68.9** | | **71.3** | 71.8 | **70.7** | | **74.8** | 74.7 | **74.3** |
| COPA | LLM.int8()♣ | | - | 86.0 | - | | - | 82.0 | - | | - | 87.0 | - | | - | 89.0 | - |
| | ZeroQuant♣ | 86.0 | 63.0 | - | 55.0 | 82.0 | 55.0 | - | 55.0 | 86.0 | 53.0 | - | 52.0 | 88.0 | 60.0 | - | 55.0 |
| | SmoothQuant | | 85.0 | - | 82.0 | | 83.0 | - | 75.0 | | 84.0 | - | 55.0 | | 88.0 | - | 55.0 |
| | Ours | | **85.0** | 86.0 | **85.0** | | **83.0** | 82.0 | **84.0** | | **85.0** | 86.0 | **84.0** | | **88.0** | 89.0 | **91.0** |
| StoryCloze | LLM.int8()♣ | | - | 76.3 | - | | - | 77.1 | - | | - | 77.7 | - | | - | 79.3 | - |
| | ZeroQuant♣ | 76.1 | 49.6 | - | 48.3 | 77.0 | 48.5 | - | 48.0 | 77.5 | 49.2 | - | 48.4 | 79.5 | 47.7 | - | 48.2 |
| | SmoothQuant | | **76.0** | - | 73.5 | | 76.9 | - | 61.4 | | 77.3 | - | 48.8 | | 79.1 | - | 49.8 |
| | Ours | | 75.8 | 76.0 | **75.4** | | **77.0** | 76.9 | **76.6** | | **77.3** | 76.4 | **76.6** | | **79.2** | 79.1 | **78.1** |
| Avg. | Ours | 65.5 | 65.5 | 65.5 | 64.7 | 67.0 | 66.9 | 66.8 | 66.7 | 68.8 | 68.5 | 68.6 | 68.0 | 71.1 | 71.0 | 71.1 | 71.1 |

Table 9: Comparison among different techniques in terms of accuracy on eight zero-shot tasks. ♣ denotes dynamic and fine-grained quantization, bringing extra computation overhead. INT8* specifically adopts per-tensor quantization for weights compared to INT8.

posed method is applied here. After obtaining a more quantization-friendly model, the MinMax algorithm collects distribution statistics. Since diverse tokens do not have outliers of varying degrees on these models, advanced clipping techniques are not involved.

**BLOOM and BLOOMZ.** The main pipeline is similar to OPTs. The only exception is using the Token-Wise Clipping as the calibration method because these models hold different outliers among different tokens. The clipping ratios are searched as 0.5% and 1.5% for 8-bit and 6-bit BLOOM, and 0.0% and 0.5% on BLOOMZ.

**LLaMA.** The main pipeline is similar to OPTs with some small differences. First, we use the Wiki-Text2 dataset for calibration. Second, as LLaMA does not have biases, introducing channel-wise shifting might incur a little overhead. Thus, for fair comparisons, we simply omit channel-wise shifting for LLaMA here. Third, when taking the harder setting that quantizes the last layer in FFN, the channel-wise scaling is also conducted thereby updating the quantization scale of `up proj` and weight parameters of `down proj`, which does not bring computation overhead during inference. Last,

unlike OPTs, for tasks with normalized accuracy metrics, we report the normalized accuracy metric instead of the accuracy one to align the original paper (Touvron et al., 2023). This point has also been indicated in each table below.

## C.2 Baselines

We introduce the implementation details of baselines here. MinMax obtains the minimum and maximum statistics of the tensor for the quantization clipping range. Percentile (Wu et al., 2020) uses the activation distribution percentile as the quantization clipping range. Using the dev set, we search its hyper-parameters within [0.999, 0.9999, 0.99999]. OMSE (Choukroun et al., 2019) minimizes the mean squared error between quantization and FP signals. PEG (Bondarenko et al., 2021) applies fine-grained quantization to problematic activation from a channel perspective. Outlier Suppression (OS) (Wei et al., 2022b) uses fixed scaling factors to suppress outliers and further clips outliers in a token-wise manner. Zero-Quant (Yao et al., 2022) uses per-token quantization, assigning different quantization parameters to different tokens. This fine-grained scheme from

| Model | Method | PIQA (↑) | | | ARC-e (↑) | | | ARC-c (↑) | | | HellaSwag (↑) | | | Winogrande (↑) | | | WikiText2 (↓) | | |
|---|---|---|---|---|---|---|---|---|---|---|---|---|---|---|---|---|---|---|---|
| | | FP16 | INT6 | INT4 | FP16 | INT6 | INT4 | FP16 | INT6 | INT4 | FP16 | INT6 | INT4 | FP16 | INT6 | INT4 | FP16 | INT6 | INT4 |
| 7B | MinMax | | **77.53** | 53.37 | | 52.36 | 29.88 | | 40.35 | 25.09 | | 70.98 | 30.98 | | 64.72 | 52.01 | | 6.22 | 430.33 |
| | SQ | 77.37 | 76.65 | 49.80 | 52.48 | **53.11** | 30.40 | 41.38 | 40.10 | 25.80 | 72.99 | 71.52 | 27.40 | 66.93 | 61.88 | 48.00 | 5.68 | 6.15 | 52.85 |
| | OS+ | | 77.20 | **64.85** | | 52.27 | **39.60** | | **40.78** | **31.06** | | **71.68** | **48.99** | | **65.11** | **54.85** | | **5.90** | **40.32** |
| 13B | MinMax | | 77.42 | 51.14 | | 57.66 | 27.61 | | 42.75 | 26.28 | | 74.72 | 25.92 | | 65.75 | 49.88 | | 5.76 | 1558 |
| | SQ | 79.05 | 77.80 | 55.55 | 59.84 | 56.36 | 34.51 | 44.62 | 42.58 | 26.71 | 76.22 | **75.11** | 41.56 | 70.09 | 68.11 | 48.70 | 5.09 | 5.50 | 79.35 |
| | OS+ | | **78.24** | **62.62** | | **57.83** | **37.67** | | **43.43** | **30.46** | | 74.96 | **52.21** | | **68.59** | **51.07** | | **5.37** | 53.64 |
| 30B | MinMax | | 74.92 | 49.46 | | 56.31 | 26.30 | | 43.69 | 29.18 | | 76.14 | 25.60 | | 69.69 | 48.62 | | 5.54 | 4958 |
| | SQ | 80.09 | 77.14 | 50.16 | 58.92 | 57.61 | 28.11 | 45.39 | 42.91 | 26.71 | 79.21 | 78.07 | 31.97 | 72.77 | 69.92 | 51.14 | 4.10 | 5.37 | 399.65 |
| | OS+ | | **79.16** | **67.19** | | **59.13** | **48.48** | | **46.25** | **35.58** | | **78.19** | **56.44** | | **72.53** | **51.85** | | **4.48** | 112.33 |
| 65B | MinMax | | 77.58 | 49.95 | | 55.18 | 26.39 | | 45.56 | 26.79 | | 78.36 | 25.35 | | 69.3 | 48.78 | | 5.98 | 54035 |
| | SQ | 80.85 | 77.97 | 61.81 | 58.75 | 54.67 | 40.15 | 46.25 | 44.62 | 32.08 | 80.73 | 77.51 | 46.19 | 77.11 | 72.61 | 50.83 | 3.56 | 4.00 | 112.02 |
| | OS+ | | **79.76** | **71.06** | | **56.31** | **49.49** | | 44.37 | **37.12** | | **79.00** | **58.76** | | **73.48** | **53.08** | | **3.82** | **32.60** |

Table 10: Comparison on LLaMA-1 in terms of normalized accuracy (Touvron et al., 2023) for the first four tasks, accuracy for Winogrande and perplexity for WikiText2. The technique in each row is equipped with per-token quantization in ZeroQuant (Yao et al., 2022). This table would quantize the last layer in FFN compared to Table 3.

the token aspect also requires dynamic quantization. Meanwhile, for INT8*, we implement per-group weight quantization according to its description. SmoothQuant (Xiao et al., 2022) migrates scaling factors to later modules to smooth problematic activation. Their scaling factors equal the range between activation and weights. For lower bits, we also search its hyper-parameter $\alpha$ according to its description for better performance.