# OpenReview forum: "Outlier Suppression+: Accurate quantization of large language models by equivalent and effective shifting and scaling"
_EMNLP/2023/Conference — EMNLP 2023 Main_

### Official Review · Reviewer_NJj5 · 2023-08-08

**Soundness:** 2

**Excitement:**

3: Ambivalent: It has merits (e.g., it reports state-of-the-art results, the idea is nice), but there are key weaknesses (e.g., it describes incremental work), and it can significantly benefit from another round of revision. However, I won't object to accepting it if my co-reviewers champion it.

**Paper Topic And Main Contributions:**

This paper proposes a new approach to build more quantization-friendly models by suppressing the outliers . With the grid-search technique to determine the optimal shifting and scaling values, the authors eliminate asymmetricity and scale of problematic channels, so that they can incur minimal quantization performance degradation, especially when bitwidth goes below 8-bit. The work have also evaluated the method on various networks and quantization settings to prove the efficacy.

**Questions For The Authors:**

- As you mentioned in section 3.2.2 and 4.5, the scaling factors learning process is unstable. The performance is poor if normal channels are scaled up.  Is it possible to modify the optimization objectives to improve the general robustness of the method?
- Can you provide more details on the implementation of 8-bit model inference in section 4.4, which demonstrates realworld acceleration when compared to FP16 models?
- How can the method  be used together with per-token quantization? The current method may have difficulty dealing with scale variance between tokens.

**Reasons To Accept:**

- The experiments are extensive .The paper evaluates the proposed method on different network sizes, tasks, and quantization settings and demonstrate that it significantly improves performance after quantization.
- The performance is good.  The 8-bit and 6-bit cases can achieves near-float point performance, and the 4-bit case is also better than other methods.

**Reasons To Reject:**

- The proposed method is like an extension of AWQ. However AWQ has not been even mentioned in the work.
- The generalization of the method needs to be expanded. For example, the LayerNorm operations in llama-like models don't have beta parameters.

**Reproducibility:**

3: Could reproduce the results with some difficulty. The settings of parameters are underspecified or subjectively determined; the training/evaluation data are not widely available.

**Reviewer Confidence:**

3: Pretty sure, but there's a chance I missed something. Although I have a good feel for this area in general, I did not carefully check the paper's details, e.g., the math, experimental design, or novelty.

---

> ### Author Rebuttal · Authors · 2023-08-29
>
> We would like to thank the reviewer for the valuable suggestions and thoughtful insight on this paper. The detailed response is listed below. Hope our reply can address the concerns.
> * **Q1**: The proposed method is like an extension of AWQ. However AWQ has not been even mentioned in the work.
>
>   **A1**: We’ll cite AWQ and discuss it in the revision. However, we’d like to first point out that we finished the submitted version of our paper in April while AWQ was made public in June. Thus, we did not incorporate any ideas from AWQ.
>
>   Furthermore, in our comparison with AWQ, there are key distinctions. First, we address different challenges: we focus on activation quantization, while they tackle weight-only quantization. Second, there are some differences between our designs. Our channel-wise shifting approach, aimed at addressing asymmetric outliers across channels, is unique and not present in AWQ. Similarly, for channel-wise scaling, our objectives (Eq.(7)) and the methods for searching scaling values (Eq.(8)) differ from theirs due to our distinct goals.
>
> * **A2**: The generalization of the method needs to be expanded. For example, the LayerNorm operations in llama-like models don't have beta parameters.
>
>   **Q2**: Thanks for your suggestion. We've observed that LLaMA models exhibit outliers with concentration and asymmetry, similar to OPTs. We've devised two ways to apply our methods to LLaMA models, as shown in the table below for LLaMA-7B.
>
>     * First, in cases where beta parameters are absent, we can utilize the scaling operation alone. As shown in the second row, this approach can bring 8-bit results close to FP16 and even improves 6-bit performance by 20 PPL.
>
>     * Second, another way is to modify the model by introducing beta and following bias parameters to support both the scaling and shifting operations. While this will increase computation overhead, we find it to be manageable. For instance, the pre-trained language model, InternLM, incorporates bias with LLaMA and introduces only a minor 5% latency increase on the 7B model and even less on larger models. Besides,  the shifting operation can enhance performance, especially on more challenging tasks like 6-bit quantization.
>
>
>     | Method             | bit-width | PIQA (acc) | WikiText (PPL) | Winogrande (acc) |
>     | ------------------ | --------- | ---------- | -------------- | ---------------- |
>     |                    | FP16      | 78.29      | 9.45           | 66.93            |
>     | MinMax             | 8-bit     | 76.88      | 10.50          | 63.77            |
>     | scaling 					|8-bit     | **78.35**  | **9.47**       | 66.93        |
>     | shifting + scaling | 8-bit     | 77.80  | **9.47**       | **67.01**        |
>     | MinMax             | 6-bit     | 67.46      | 31.00          | 55.94            |
>     | scaling            | 6-bit     | 75.73      | 10.88          | 62.04            |
>     | shifting + scaling | 6-bit     | **77.26**  | **10.01**      | **65.75**        |
>
> * **Q3**: As you mentioned in section 3.2.2 and 4.5, the scaling factors learning process is unstable. The performance is poor if normal channels are scaled up. Is it possible to modify the optimization objectives to improve the general robustness of the method?
>
>   **A3**: Thanks for your suggestion. We'd like to first clarify that our method does not directly learn the scaling factors but is a way to modify the optimization objective to efficiently and stably obtain scaling values. Recognizing that normal values often vary across samples, our approach introduces an outlier threshold in Eq. (8) to represent scaling values. Then, we inject Eq. (8) into the objective Eq. (6) and Eq. (7). Optimization in such a way scales down outlier channels while keeping normal values intact. Second, other ways to modify the objective are also worth exploring like adding a penalty term of scaling values in the objective.
>
> * **Q4**: Can you provide more details on the implementation of 8-bit model inference in section 4.4, which demonstrates realworld acceleration when compared to FP16 models?
>
>   **A4**: Thanks for your suggestion. We will provide the entire procedure from exporting improved FP models with our method to latency testing using FasterTransformer. Here are the key points:
>
>     * For LLMs, our methods only update the weights and biases of FP models, ensuring compatibility with common frameworks. Specifically, we employ FasterTransformer to directly measure the latency of 8-bit quantization using our more quantization-friendly FP models. Importantly, this design does not introduce any additional computational burden during inference.
>
>    * For BERT models, in pursuit of FP equivalence in Post-LN structure, we implement structural modifications within FasterTransformer, which replaces the shortcut branch with channel-wise addition and multiplication.  With our enhanced FP model, we measure its latency, as illustrated in Figure 4. Notably, these structural changes result in a negligible increase in latency.
>
>
> * **Q5**: How can the method be used together with per-token quantization? The current method may have difficulty dealing with scale variance between tokens.
>
>   **A5**: Our methods focus on addressing outliers in the channel dimension and can be used in conjunction with per-token quantization. The shifting operation reduces the tensor range to the maximum channel range, while the scaling operation scales down outlier channels for all tokens. Based on our more quantization-friendly FP model, per-token quantization can then be applied to deal with token variance for better quantization results.
>
>   We conduct experiments on a harder task: 5-bit quantization, and show results on OPT-66B. It can be seen that ours and per-token quantization can help each other to pursue favorable results.
>
>   |                  | PIQA  | Winogrande |
>   | :--------------- | ----- | ---------- |
>   | FP16             | 78.73 | 68.90      |
>   | per-token        | 51.90 | 50.20      |
>   | ours             | 76.50  | 67.56       |
>   | per-token + ours | **78.07**  | **68.43**      |

---

### Official Review · Reviewer_rLws · 2023-08-09

**Soundness:** 4

**Excitement:**

3: Ambivalent: It has merits (e.g., it reports state-of-the-art results, the idea is nice), but there are key weaknesses (e.g., it describes incremental work), and it can significantly benefit from another round of revision. However, I won't object to accepting it if my co-reviewers champion it.

**Missing References:**

N/A

**Paper Topic And Main Contributions:**

This work proposes a quantization framework Outlier Suppression+ for LLMs. The target is to apply this framework to improve the efficiency of LLMs on various hardware systems with minimum computation overhead. The main contribution of this paper is first to analyze the outlier characteristics of LLMs. The channel-wise shifting and scaling operations are utilized to suppress the outliers. Outliers are further scaled down by balancing the burden between weight and activation quantization. The method is evaluated across different models (BERT, OPTs, BLOOM) and various tasks (GLUE, PIQA, HellaSwag, etc.) to demonstrate effectiveness and efficiency.

**Questions For The Authors:**

A. What is the core novelty of the proposed channel-wise shifting and channel-wise scaling?
B. In Table 2, is INT8* indicate per-tensor quantization while INT8 represents per-token quantization? Why does applying per-tensor quantization for weights (which is a coarser granularity compared to per-token) even achieve better performance with the proposed method?
C. Can this method apply to LLaMa?

**Reasons To Accept:**

1. The motivation of this work is clear and reasonable. Some insights on eliminating the outliers are given and are helpful for future research on LLMs.
2. The experiment design is comprehensive, convering both lightweight LMs and LLMs.
3. Real latency on hardware is provided, strengthening the results' credibility.
4. The presentation quality of this paper is good, and the content is easy-to-follow.

**Reasons To Reject:**

1. The novelty of the proposed method (especially channel-wise shifting and channel-wise scaling) are questionable. The channel-wise shifting is to some extent similar to a standard normalization and is widely used in previous symmetry quantization methods. To balance the burden between weight and activation quantization is also a core contribution of the previous method SmoothQuant[1]. More discussion should be included in the revision.

2. The performance improvement of INT8 quantization on zero-shot tasks is limited (compared to SmoothQuant)

3. Figure 5 is not a straightforward visualization as the storage has been the x-axis, it is redundant to draw some circles/rectangles to use area to represent it.

[1] SmoothQuant: Accurate and Efficient Post-Training Quantization for Large Language Models, ICML 2023

**Reproducibility:**

4: Could mostly reproduce the results, but there may be some variation because of sample variance or minor variations in their interpretation of the protocol or method.

**Reviewer Confidence:**

4: Quite sure. I tried to check the important points carefully. It's unlikely, though conceivable, that I missed something that should affect my ratings.

**Typos Grammar Style And Presentation Improvements:**

Format: The submission has exceeded the page limits of most 8 pages.

---

> ### Author Rebuttal · Authors · 2023-08-29
>
> We would like to sincerely thank the reviewer for providing insightful suggestions on this paper. The detailed response is listed below. We hope our reply can address the questions.
> * **Q1** : The novelty of the proposed method (especially channel-wise shifting and channel-wise scaling) are questionable. The channel-wise shifting is to some extent similar to a standard normalization and is widely used in previous symmetry quantization methods. To balance the burden between weight and activation quantization is also a core contribution of the previous method SmoothQuant[1]. More discussion should be included in the revision.
>
>   **A1**: Thanks for the comment. We’d like to clarify our novelty in the following and will include more discussion in the revision.
>
>     * Channel-wise shifting: Our shifting operation aims to improve per-tensor activation quantization rather than symmetric quantization. We observed that outliers tend to be either extremely positive or negative within different channels. To address this, we align channel centers to reduce the entire tensor range to its maximum channel range. This approach is distinct from standard normalization, which, to our best knowledge, refers to weight standardization for regularization or distribution normalization of weights only during quantization-aware training. Additionally, our designed shifting contains a migration pattern to ensure FP equivalence in a post-training setting.
>
>     * Channel-wise scaling: Inspired by Outlier Suppression [1] which was the first to propose activation scaling in outlier quantization, we share the same goal with SmoothQuant [2] in addressing outlier concentration. However, our method distinguishes them by proposing a more rational approach to calculating scaling values, resulting in significant performance gains, such as a 17.4% improvement on Winogrande compared to [2]. Unlike [1] which reduces quantization error for individual tensors or [2] which equalizes weight and activation ranges, our method proposes to optimize their interactive outputs. We employ a quantitative method with a fast search process to minimize output changes.
>
>   [1]. Outlier Suppression: Pushing the Limit of Low-bit Transformer Language Models. NeurIPS2022
>
>   [2]. SmoothQuant: Accurate and Efficient Post-Training Quantization for Large Language Models, ICML 2023
>
> * **Q2**: The performance improvement of INT8 quantization on zero-shot tasks is limited (compared to SmoothQuant)
>
>   **A2**: For 8-bit quantization on OPT, both our method and SmoothQuant approach the FP16 baseline. However, it should be highlighted that our results on INT8 BLOOM close **60%** of the gap between FP16 baseline and SmoothQuant INT8 (e.g., SmoothQuant: 67.4, Ours: 67.9, FP16: 68.2). In our view, the results are still significant.
>
>   On the more challenging 6-bit quantization task, our method exhibits even more pronounced improvements, with a 28.3% enhancement on the StoryCloze task and a 27.4% boost on PIQA.
>
>
> * **Q3**: Figure 5 is not a straightforward visualization as the storage has been the x-axis, it is redundant to draw some circles/rectangles to use area to represent it.
>
>   **A3**: Thanks for your suggestion. In the revision, we will use different marks with consistent areas to represent different bit-widths.
>
> * **Q4**: A. What is the core novelty of the proposed channel-wise shifting and channel-wise scaling?
>
>   **A4**: We conclude our contribution in the novelty aspect in the following:
>
>     * The channel-wise shifting finds a new characteristic of outliers – their asymmetry across channels, which can significantly expand the tensor range, even with a small channel range. To mitigate this, we introduce the shifting operation to enhance per-tensor quantization while preserving FP equivalence through a migration pattern.
>
>     * The channel-wise scaling addresses outlier concentration with an innovative approach for scaling values. Instead of reducing quantization errors for individual tensors or using heuristic methods, it proposes to minimize interactive output changes of weights and activations. This optimization is achieved through a fast and stable search process, resulting in significant improvements, such as a 17.4% gain compared to SmoothQuant.
>
> * **Q5**: In Table 2, is INT8* indicate per-tensor quantization while INT8 represents per-token quantization? Why does applying per-tensor quantization for weights (which is a coarser granularity compared to per-token) even achieve better performance with the proposed method?
>
>   **A5** : INT8* indicates per-tensor quantization and INT8 only changes the weight quantization to the per-channel scheme. Note that activation always takes the per-tensor quantization but both per-tensor and per-channel schemes are common for weights.
>
>   For the better performance brought by coarser granularity, we guess the reviewer refers to cases like PIQA on OPT-13B, where INT8* achieves 76.4% accuracy, compared to 75.9% for INT8. We think this is because they have been close to the FP16 baseline at 75.8, and the PIQA datasets with limited size might make it challenging to distinguish these subtle differences among good enough results.
>
>   However, the last row in Table 2 shows the averaged results across tasks. It can be seen that per-tensor quantization only surpasses per-channel (weight) quantization by 0.1% on OPT-30B. On others, INT8 performs equally or better than INT8*.
>
> * **Q6**: Can this method apply to LLaMa?
>
>   **A6**: Our method is applicable to LLaMA, where we also observe outliers concentrated in specific channels (e.g., the 3968 and 3180 channels in one layer) and asymmetric shape across channels (e.g., (1.0, 17.7) range on the 3968 channel and (-27.3, -19.6) on 3180). Therefore, our methods effectively suppress these outliers.
>
>    In the table below, we conduct experiments on LLaMa-7B and we will include more complete results in the revision. For 8-bit and 6-bit quantization, our methods can bring close results to FP16. Especially, on 6-bit, the scaling operation can even improve 20 PPL and the shifting operation further brings a 2~3% boost on PIQA and Winogrande by aligning distribution across channels.
>
>     | Method             | bit-width | PIQA (acc) | WikiText (PPL) | Winogrande (acc) |
>     | ------------------ | --------- | ---------- | -------------- | ---------------- |
>     |                    | FP16      | 78.29      | 9.45           | 66.93            |
>     | MinMax             | 8-bit     | 76.88      | 10.50          | 63.77            |
>     | shifting + scaling | 8-bit     | **77.80**  | **9.47**       | **67.01**        |
>     | MinMax             | 6-bit     | 67.46      | 31.00          | 55.94            |
>     | scaling            | 6-bit     | 75.73      | 10.88          | 62.04            |
>     | shifting + scaling | 6-bit     | **77.26**  | **10.01**      | **65.75**        |
>
> * **Q7**: Format: The submission has exceeded the page limits of most 8 pages.
>
>   **A7**: Thanks for the comment. We find that the limitation section is not included in the page limit. We will put the whole limitation section on page 9 for better presentation in the revision.

---

### Official Review · Reviewer_vshF · 2023-08-09

**Soundness:** 4

**Excitement:**

4: Strong: This paper deepens the understanding of some phenomenon or lowers the barriers to an existing research direction.

**Paper Topic And Main Contributions:**

In this study, an efficient method to suppress outliers in the activation of large transformer language models for Quantization is proposed : Channel-wise shifting and scaling operations. These techniques can eliminate assymmetry in the distribution of activations in each channel and scale down the channels that have outliers. Moreover, the operations can be almost absorbed into the existing architectures by a simple correction of the parameters, so as not to burden the computational complexity. They also show the experimental results on BERT and LLMs (OPTs, BLOOM, BLOOMz with sizes ranging from 13B to 176B), the accuracy degradation of the proposed method is comparable to the recent similar approach, SmoothQuant in per-tensor INT8 quantization, and better than that in per-channel INT6 quantization.

**Questions For The Authors:**

It is necessary to tell this method from SmoothQuant and AWQ in detail.

**Reasons To Accept:**

The simple methods can suppress outliers in activations and successfully quantize LLMs without degrading accuracy. The operations can almost be absorbed into the existing architectures so as not to burden the computational complexity. The idea is similar to SmoothQuant, but this method outperforms it in per-channel INT6 quantization. The quantization algorithm is efficient in that the computation of shifting and scaling values takes only 20 minutes for 175B LLMs, and the real latency shows a 1.5x speed-up compared to FP16.

**Reasons To Reject:**

The idea is similar to that of SmoothQuant and AWQ. Of course, these ideas could have been developed independently, but it's clear that SmoothQuant precedes it.

**Reproducibility:**

3: Could reproduce the results with some difficulty. The settings of parameters are underspecified or subjectively determined; the training/evaluation data are not widely available.

**Reviewer Confidence:**

3: Pretty sure, but there's a chance I missed something. Although I have a good feel for this area in general, I did not carefully check the paper's details, e.g., the math, experimental design, or novelty.

---

> ### Author Rebuttal · Authors · 2023-08-28
>
> We thank the reviewer for the constructive comments and feedback. Hope the following reply helps address the concerns.
>
> * **Q1** : The idea is similar to that of SmoothQuant and AWQ. Of course, these ideas could have been developed independently, but it's clear that SmoothQuant precedes it.
>
>   **A1**: Our framework is inspired by the idea from Outlier Suppression [1], which was the first to introduce activation scaling to alleviate outlier quantization in language models. As we have stated the difference with it in the paper clearly, we’d like to highlight our differences compared to SmoothQuant and AWQ in the following:
>
>     * Compared to SmoothQuant: First, we identify a new feature of outliers that they stay in **asymmetric shape across channels**. To eliminate such a phenomenon, we propose the shifting operation to align channels for better per-tensor quantization with a migration pipeline for FP equivalence. The finding and technique did not appear in SmoothQuant.
>
>       Second, for the outlier concentration phenomenon, we propose a different and better solution, which can provide large improvements such as a **17.4% performance boost** on Winogrande. Considering scaling values act in both the activation and weights, we notice that reducing quantization error of individual tensors is sub-optimal and propose to optimize towards their interactive output change. Our scaling operation uses a fast and stable search pipeline to quantitatively evaluate the objective. However, SmoothQuant adopts a heuristic way which equalizes ranges between activation and weights and can not guarantee the minimum output change.
>
>    * Compared to AWQ: First, we completed our submitted paper in **April**, while AWQ was made public in **June**, precluding any opportunity for us to integrate ideas from it. Second, our work and that of AWQ address different problems and employ distinct methods. We concentrate on the challenging activation quantization, whereas they focus on weight-only quantization. Additionally, our introduction of the shifting operation to tackle asymmetric outliers across channels is unique and not present in theirs. Given our different problems, our scaling values do not take the same objective (Eq. (7)) as theirs and employ a different search method (Eq. (8)).
>
>
>    [1]. Outlier Suppression: Pushing the Limit of Low-bit Transformer Language Models. NeurIPS2022.
>
> * **Q2**: It is necessary to tell this method from SmoothQuant and AWQ in detail.
>
>   **A2**: Thanks for your suggestion. We will discuss differences between SmoothQuant, AWQ and us clearly in the revision to avoid any potential misunderstanding.

---

### Official Review · Reviewer_uJMf · 2023-08-10

**Typos Grammar Style And Presentation Improvements:** 1. In Figure 2, the colors of the ele…
**Soundness:** 4

**Excitement:**

4: Strong: This paper deepens the understanding of some phenomenon or lowers the barriers to an existing research direction.

**Paper Topic And Main Contributions:**

This paper presents a method for normalizing outlier channels to mitigate the widespread outlier phenomenon in LLMs. The shift amplitude is determined by the minimum and maximum values in the collaboration datasets, while the scale value is obtained through module-level MSE loss search. The authors tested three model families: BERTs, OPTs, and BLOOMs, evaluating their performance in text classification tasks and zero-shot QA tasks using three different bit levels: INT8, INT6, and INT4. The experimental results show a significant performance improvement over the state-of-the-art LLMs quantitative methods at the INT6 bit level, and the performance loss compared to FP16 is relatively small.

**Questions For The Authors:**

**Question 1**: Is there any evidence to suggest that this outlier asymmetric distribution is static and can be estimated from a small number of samples? You could consider searching for z and s within the in-distribution dataset and then directly testing on the out-of-distribution datasets.

**Question 2**: Have you tested the method on tasks that are more sensitive to parameter changes, such as language modeling tasks like Wikitext?

**Question 3**: Have you tested more aggressive bits like 4-bit or 5-bit on OPTs or BLOOMZs?


**Reasons To Accept:**

1. The motivation of the proposed approach is sound.
2. The paper clearly discusses the rationale behind the outlier shifting and scaling, as well as the parameter calculation design, and provides easily understandable detailed experiments for its design choices.
3. Without outlier shifting and scaling, Tables 2 and 5 show the performance drop of previous state-of-the-art methods at lower bit levels, such as INT6. Moreover, using OS+ enables LLMs to achieve almost no performance loss in zero-shot QA tasks.

**Reasons To Reject:**

1. While the authors claim that z and s can be determined with only a small number of samples, it is difficult not to question whether these few samples can provide sufficient diversity and completeness for the distribution boundary.
2. Although the results in Section 4 are quite good, QA and text classification tasks inherently have a higher tolerance for errors. It is uncertain how well the method would perform on tasks that are more sensitive to parameter changes, such as language modeling tasks.



**Reproducibility:**

4: Could mostly reproduce the results, but there may be some variation because of sample variance or minor variations in their interpretation of the protocol or method.

**Reviewer Confidence:**

4: Quite sure. I tried to check the important points carefully. It's unlikely, though conceivable, that I missed something that should affect my ratings.

---

> ### Author Rebuttal · Authors · 2023-08-28
>
> Thanks for the reviewer’s valuable suggestions and positive feedback on this paper. Below is the detailed response to each question. Hope the following reply helps address the concerns.
>
> * **Q1** : While the authors claim that z and s can be determined with only a small number of samples, it is difficult not to question whether these few samples can provide sufficient diversity and completeness for the distribution boundary.
>
>   **A1** : Thanks for the question. We find that outliers in transformer language models show some static features including concentration on fixed channels [1, 2, 3] and similar extreme positive and negative values across batches of samples, indicating that a small calibration size for getting s and z is enough.
>
>   To verify this, we provide an ablation study (which will be included in the revised draft). The calibration samples are randomly chosen from the PILE dataset (one of the train datasets). From the second column, it shows that sample sets of different sizes have similar ranges.
>
>    | Number of samples | min range /max range of one layer |
>    | ----------------- | --------------------------------- |
>    | 16                | -93.8750 / 31.5938                |
>    | 64                | -93.8750/ 31.7656                 |
>    | 96                | -94.0000 / 31.8125                |
>    |128               | -94.0000 / 31.8125                |
>
>    Besides, we give a possible explanation by attributing the observation to the inherent behaviors of per-trained models. Like [2], we also observe that the boundaries are often provided by high-frequency tokens like the separator one. Thus, going through batches of samples with an output length set to 512 can include high-frequency tokens and is enough for the boundaries.
>
>   [1]. Understanding and Overcoming the Challenges of Efficient Transformer Quantization. EMNLP2021
>
>   [2]. Outlier Suppression: Pushing the Limit of Low-bit Transformer Language Models. NeurIPS2022
>
>   [3]. LLM.int8(): 8-bit Matrix Multiplication for Transformers at Scale. NeurIPS2022
>
> * **Q2**: Although the results in Section 4 are quite good, QA and text classification tasks inherently have a higher tolerance for errors. It is uncertain how well the method would perform on tasks that are more sensitive to parameter changes, such as language modeling tasks.
>
>   **A2**: Thanks for your suggestion. We conduct experiments on WikiText here. It can be seen that our method still achieves near-floating point performance for these models on 8-bit and 6-bit quantization. This indicates that our scaling and shifting values found by PILE dataset are good enough.
>
>
>     | WikiText (PPL)     | OPT-13B | OPT-30B | OPT-66B | OPT-175B |
>     |--------------------|---------|---------|---------|----------|
>     | FP16               | 14.13   | 13.09   | 12.30   | 11.00    |
>     | 8-bit              | 14.21   | 13.15   | 12.35   | 11.02    |
>     | 6-bit              | 14.41   | 13.34   | 12.49   | 11.13    |
>
> * **Q3**: Is there any evidence to suggest that this outlier asymmetric distribution is static and can be estimated from a small number of samples? You could consider searching for z and s within the in-distribution dataset and then directly testing on the out-of-distribution datasets.
>
>   **A3**: Yes, in the table for Q1 here, the second column shows close boundaries across different calibration sizes. For experiments of LLMs, to evaluate zero-shot tasks, we adopted a different dataset PILE (one of the train datasets) for calculating s and z and quantization calibration. Then, experiments are conducted directly on the out-of-distribution datasets including LAMBADA, PIQA, Winogrande, etc. Good results in the paper indicate the superiority of parameters determined by the in-distribution dataset and suggest static features of outliers.
>
> * **Q4**: Have you tested the method on tasks that are more sensitive to parameter changes, such as language modeling tasks like Wikitext?
>
>   **A4**: Yes,  we have tested on the LAMBADA dataset with PPL and accuracy metrics. Due to the space limit, we did not report the PPL one in the original paper but will include them in the revision. Also, we conduct experiments on WikiText in the table for Q2 here.
>
> * **Q5**: Have you tested more aggressive bits like 4-bit or 5-bit on OPTs or BLOOMZs?
>
>   **A5**: Thanks for the insightful comment. We find that 4-bit quantization is much more challenging with limited bit levels to represent too many values. For example, 4-bit weight-only quantization can incur an accuracy drop [1].
>
>     However, motivated by ZeroQuant-V2 which adopts per-group quantization for W4A8, we also combine the per-group quantization with our method and report results on W4A4 with group sizes 1024 and 512.
>
>
>     | 1024 group size     | OPT-13B   | OPT-30B   | OPT-66B   | OPT-175B  |
>     | ------------------- | --------- | --------- | --------- | --------- |
>     | **WikiText (FP16)** | 14.13     | 13.09     | 12.30     | 11.00     |
>     | MinMax              | 17011.67  | 44.89     | 239.02    | 26.91     |
>     | Ours                | **15.48** | **13.89** | **12.90** | **11.24** |
>     | **PIQA (FP16)**     | 75.79     | 77.58     | 78.73     | 79.71     |
>     | MinMax              | 53.21     | 67.63     | 57.67     | 64.80     |
>     | Ours                | **74.16** | **76.88** | **77.69** | **79.82** |
>
>     | 512 group size      | OPT-13B   | OPT-30B   | OPT-66B   | OPT-175B  |
>     | ------------------- | --------- | --------- | --------- | --------- |
>     | **WikiText (FP16)** | 14.13     | 13.09     | 12.30     | 11.00     |
>     | MinMax              | 1149.70   | 19.52     | 18.57     | 15.27     |
>     | Ours                | **14.50** | **13.43** | **12.66** | **11.18** |
>     | **PIQA (FP16)**     | 75.79     | 77.58     | 78.73     | 79.71     |
>     | MinMax              | 62.24     | 72.74     | 74.59     | 75.75     |
>     | Ours                | **74.96** | **77.20** | **77.97** | **79.71** |
>
>    It can be seen that ours can achieve near-floating point performance even with large  group size, and can surpass MinMax by a large margin. Especially, it can be observed that the number of groups can be decreased with our methods to reach satisfying results.  We will include more 4-bit quantization experiments in the revision.
>
>
>    [1]. GPTQ: Accurate Post-Training Quantization for Generative Pre-trained Transformers. ICLR2023.
>
>    [2]. ZeroQuant-V2: Exploring Post-training Quantization in LLMs from Comprehensive Study to Low Rank Compensation.
>
> * **Q6**: Typos Grammar Style And Presentation Improvements.
>
>   **A6**: Thanks for pointing these out. We will fix the color and baseline parts in the revision.

---

### Official Review · Reviewer_AfNB · 2023-08-13

**Soundness:** 3

**Excitement:**

3: Ambivalent: It has merits (e.g., it reports state-of-the-art results, the idea is nice), but there are key weaknesses (e.g., it describes incremental work), and it can significantly benefit from another round of revision. However, I won't object to accepting it if my co-reviewers champion it.

**Paper Topic And Main Contributions:**

This paper presents a novel approach to addressing the challenges posed by outliers in the quantization of transformer language models. Taking into account the observed asymmetric characteristics of these outliers, the authors propose an efficient shifting and scaling mechanism to mold a quantization-friendly distribution. They further demonstrate that implementing this technique preserves the equivalence of the original model computations. By adapting the quantization target distribution to be more amenable to quantization, the method achieved superior and more robust performance than previous state-of-the-art methods, specifically on BERT with W4A4 and LLM with W6A6 configurations.

**Questions For The Authors:**

- Is there a specific reason for the absence of 4-bit quantization results for LLM, unlike BERT? I'm curious to know if applying shifting/scaling to INT4 quantization in LLM remains a challenging endeavor.
- Do the authors believe that the proposed activation shifting is orthogonal and can be applied in conjunction with other PTQ methods, such as SmoothQuant? It seems that it might enhance the performance of other methods as well. I'm interested to know author's perspective on this.

**Reasons To Accept:**

- The paper is well-structured overall, facilitating easy comprehension. It effectively represents the need for shifting/scaling and illustrates the process required in subsequent modules to maintain equivalence clearly.
- The paper's identification of asymmetric distribution as a challenging aspect in per-tensor activation quantization is insightful. Proposing the shifting technique to address this concern adds to its novelty.
- The authors convincingly showcase the efficiency of the proposed algorithm and its deployment efficiency, underscoring its practical applicability.

**Reasons To Reject:**

- **Generalization and Evaluation**: The paper appears to fall short in generalizing its methodology to other pre-trained GLM models. While the approach may prove potent for models like OPT where asymmetry in activation is pronounced, its effectiveness might be attenuated for GLMs that are more quantization-friendly in their activation. (This can be inferred from the performance discrepancy between OPT in INT6 and BLOOMZ in INT6). It would be beneficial to observe results on widely-used pre-trained GLMs like LLaMA to better assess the utility of this method. Additionally, it seems the paper omits a comparative analysis on the Perplexity (PPL) measure, which typically gauges a decoder model's generation ability.

- **Insufficient Analysis**: The paper appears to lack a comprehensive analysis of how the proposed shifting and scaling methods influence the reduction of quantization error. For instance, the introduction mentions that "Existing work did not quantitatively analyze the effect of activation scaling on the quantization of subsequent layer's weight well." Yet, it's unclear if this paper provides an analysis beyond performance differences concerning the effects of scaling on subsequent layers. It would be helpful to clarify if there's an aspect I might be overlooking. A suggestion would be to delve deeper into how the scaling values determined by the proposed method correlate with module-specific distributions. A more in-depth analysis in this regard could significantly bolster the paper's strength.

**Reproducibility:**

4: Could mostly reproduce the results, but there may be some variation because of sample variance or minor variations in their interpretation of the protocol or method.

**Reviewer Confidence:**

4: Quite sure. I tried to check the important points carefully. It's unlikely, though conceivable, that I missed something that should affect my ratings.

**Typos Grammar Style And Presentation Improvements:**

- It would be beneficial to condense the tasks presented in the main table and also include results for other GLM models for a comprehensive comparison.

- One of the most innovative aspects of this paper's methodology appears to be the shifting of the asymmetric distribution for per-tensor activation quantization, distinguishing it from conventional quantization-friendly approaches. It would be beneficial to emphasize this aspect more prominently.

---

> ### Author Rebuttal · Authors · 2023-08-28
>
> We sincerely thank the reviewer for the constructive comments and feedback. Hope the following reply helps address the concerns.
> * **Q1**: Generalization and Evaluation: The paper appears to fall short in generalizing its methodology to other pre-trained GLM models. While the approach may prove potent for models like OPT where asymmetry in activation is pronounced, its effectiveness might be attenuated for GLMs that are more quantization-friendly in their activation. (This can be inferred from the performance discrepancy between OPT in INT6 and BLOOMZ in INT6). It would be beneficial to observe results on widely-used pre-trained GLMs like LLaMA to better assess the utility of this method. Additionally, it seems the paper omits a comparative analysis on the Perplexity (PPL) measure, which typically gauges a decoder model's generation ability.
>
>   **A1**: Thanks for the valuable suggestion.
>   * We investigate LLaMA models and find that they also hold the outlier concentration phenomenon (e.g., the 3968 and 3180 channels in one layer) and asymmetric outliers across channels (e.g., (1.0, 17.7) range on the 3968 channel and (-27.3, -19.6) on 3180). Thus, our methods can also address the similar problem for these models. The table below shows the results of LLaMA-7B.
>
>     | Method             | bit-width | PIQA (acc) | WikiText (PPL) | Winogrande (acc) |
>     | ------------------ | --------- | ---------- | -------------- | ---------------- |
>     |                    | FP16      | 78.29      | 9.45           | 66.93            |
>     | MinMax             | 8-bit     | 76.88      | 10.50          | 63.77            |
>     | shifting + scaling | 8-bit     | **77.80**  | **9.47**       | **67.01**        |
>     | MinMax             | 6-bit     | 67.46      | 31.00          | 55.94            |
>     | scaling            | 6-bit     | 75.73      | 10.88          | 62.04            |
>     | shifting + scaling | 6-bit     | **77.26**  | **10.01**      | **65.75**        |
>
>     It can be seen that ours can achieve near-floating point performance on 8-bit and 6-bit. Our scaling operation can even improve 20 PPL and the shifting operation further brings 2-3% accuracy improvement on PIQA and Winogrande. We will provide more experiments in the revision.
>   * Sorry for the omitting of the PPL metric. Due to the space limit, we only report the accuracy metric for the LAMBABA dataset in the original paper. We will include its PPL in the revision. To further address the concern, we conduct the experiment on the WikiText dataset with the PPL metric in the table below. It can be seen that our results are still close to FP16.
>     | WikiText (PPL)     | OPT-13B | OPT-30B | OPT-66B | OPT-175B |
>     |--------------------|---------|---------|---------|----------|
>     | FP16               | 14.13   | 13.09   | 12.30   | 11.00    |
>     | 8-bit              | 14.21   | 13.15   | 12.35   | 11.02    |
>     | 6-bit              | 14.41   | 13.34   | 12.49   | 11.13    |
>
> * **Q2**:  Insufficient Analysis: The paper appears to lack a comprehensive analysis of how the proposed shifting and scaling methods influence the reduction of quantization error. For instance, the introduction mentions that "Existing work did not quantitatively analyze the effect of activation scaling on the quantization of subsequent layer's weight well." Yet, it's unclear if this paper provides an analysis beyond performance differences concerning the effects of scaling on subsequent layers. It would be helpful to clarify if there's an aspect I might be overlooking. A suggestion would be to delve deeper into how the scaling values determined by the proposed method correlate with module-specific distributions. A more in-depth analysis in this regard could significantly bolster the paper's strength.
>
>    **A2**: Thanks for your suggestion. We’d like to give clearer analyses in the following:
>    * The channel-wise shifting reduces the quantization error by decreasing the tensor range to its maximum channel range. For instance, on OPT-66B, where outliers exhibit a wide range (-93.9, 31.6), the mean squared quantization error is 209.8. The shifting operation aligns the center of each channel, changing the range to **(-13.9, 13.9)** followed by a smaller quantization error of **112.8**.
>
>   * The channel-wise scaling reduces the quantization error compared to other scaling methods by quantitatively minimizing interactive output change of activation and weights. We will make the mentioned sentence clearer in the revision. Because scaling values act in both the activation and weights, reducing quantization error for individual tensors can not guarantee the minimum output change, which encapsulates their information to later forward pass.
>
>     |                       |               activation  |       |        weight   |       | Output change |
>     |-----------------------|---------------------------|-------|-----------------|-------|---------------|
>     |                       | range                     | MSE   | range           | MSE   | MSE           |
>     | original              | (-93.9, 31.6)             | 209.8 |  (-0.13, 0.13)  | 0.001 |   18061.5     |
>     | Outlier Suppression   | (-23.5,15.7)              | 142.9 | **(-0.40, 0.41)** | **0.006** |    6182.52    |
>     | SmoothQuant           | **(-3.5, 2.0)**           | **3.65** | (3.4, 3.5)      | 0.43  |    3535.86    |
>     | Our scaling operation | (-8.4, 8.4)              | 48.54 |  (1.2, 1.3)     | 0.02  |    **1334.89**    |
>
>     For example, in the table above, Outlier Suppression with fixed scaling values has the smallest quantization error for weight. SmoothQuant with a heuristic way has the smallest quantization error for activation. However, both of them did not bring the smallest quantization error for the output. This reveals the importance of directly optimizing according to the output, which is what our method exactly does. Thus, we can enjoy the best final performance.
>
> * **Q3** : Is there a specific reason for the absence of 4-bit quantization results for LLM, unlike BERT? I'm curious to know if applying shifting/scaling to INT4 quantization in LLM remains a challenging endeavor.
>
>   **A3**: Thanks for your suggestion. We did not conduct experiments on 4-bit LLMs because compared to BERT, tensors in LLMs have much more elements that makes quantization challenging. For example, 4-bit weight-only quantization incurs an accuracy drop [1].
>
>    Nevertheless, motivated by ZeroQuant-v2 [2] which adopts per-group quantization for W4A8, we also combine per-group quantization with our method and report results on W4A4 in the table below.
>
>     | 1024 group size     | OPT-13B   | OPT-30B   | OPT-66B   | OPT-175B  |
>     | ------------------- | --------- | --------- | --------- | --------- |
>     | **WikiText (FP16)** | 14.13     | 13.09     | 12.30     | 11.00     |
>     | MinMax              | 17011.67  | 44.89     | 239.02    | 26.91     |
>     | Ours                | **15.48** | **13.89** | **12.90** | **11.24** |
>     | **PIQA (FP16)**     | 75.79     | 77.58     | 78.73     | 79.71     |
>     | MinMax              | 53.21     | 67.63     | 57.67     | 64.80     |
>     | Ours                | **74.16** | **76.88** | **77.69** | **79.82** |
>
>     | 512 group size      | OPT-13B   | OPT-30B   | OPT-66B   | OPT-175B  |
>     | ------------------- | --------- | --------- | --------- | --------- |
>     | **WikiText (FP16)** | 14.13     | 13.09     | 12.30     | 11.00     |
>     | MinMax              | 1149.70   | 19.52     | 18.57     | 15.27     |
>     | Ours                | **14.50** | **13.43** | **12.66** | **11.18** |
>     | **PIQA (FP16)**     | 75.79     | 77.58     | 78.73     | 79.71     |
>     | MinMax              | 62.24     | 72.74     | 74.59     | 75.75     |
>     | Ours                | **74.96** | **77.20** | **77.97** | **79.71** |
>
>     It can be seen that ours can achieve near-floating point performance even with a big group size and can surpass MinMax by a large margin. Especially, it can be observed that the number of groups can be decreased with our methods to reach satisfying results. We will include more results on 4-bit quantization in the revision.
>
>    [1]. GPTQ: Accurate Post-Training Quantization for Generative Pre-trained Transformers. ICLR2023
>
>    [2]. ZeroQuant-V2: Exploring Post-training Quantization in LLMs from Comprehensive Study to Low Rank Compensation.
>
>  * **Q4**: Do the authors believe that the proposed activation shifting is orthogonal and can be applied in conjunction with other PTQ methods, such as SmoothQuant? It seems that it might enhance the performance of other methods as well. I'm interested to know author's perspective on this.
>
>    **A4**: Yes, we agree with the reviewer’s point that the channel-wise shifting is orthogonal. First, it works from a different observation about outliers that they show asymmetric shape across channels. Second, it makes the activation more friendly for per-tensor quantization, giving better support to subsequent techniques. In the table below, we combine it with SmoothQuant on OPT-66B and Winogrande.
>
>    |              | SmoothQuant | Our scaling operation |
>    | ------------ | ----------- | --------------------- |
>    | w/o shifting | 51.14       | **66.5**              |
>    | w/ shifting  | **66.69**   | **69.4**              |
>
>    It can be seen that our shifting operation can boost the performance of SmoothQuant from 51.14 to 66.69 with **15.54%** improvement.  Also, considering that our scaling operation is better than that of SmoothQuant (**66.5 vs 51.14**), combining both of our operations can finally achieve a near floating-point performance.
>
> * **Q5**: Typos Grammar Style And Presentation Improvements.
>
>   **A5**: Thanks for the suggestions. We will condense the main table to include more pre-trained models, and emphasize more about the channel-wise shifting.

---

### Meta-Review · Area_Chair_n8Tx · 2023-09-07

**Recommendation:** 5

**Metareview:**

Reviewers almost unanimously recommend acceptance with strong soundness and moderate excitement. The experimental setup and results are rigorous, the method well-motivated, and the results are strong. I recommend acceptance to the main conference.

---

### Decision · Program_Chairs · 2023-10-07

**Decision:**

Accept-Main

**Comment:**

Reviewers almost unanimously recommend acceptance with strong soundness and moderate excitement. The experimental setup and results are rigorous, the method well-motivated, and the results are strong. I recommend acceptance to the main conference.